# CONSTRAINED PARAMETER REGULARIZATION

## ABSTRACT

Regularization is a critical component in deep learning training, with weight decay being a commonly used approach. It applies a constant penalty coefficient uniformly across all parameters. This may be unnecessarily restrictive for some parameters, while insufficiently restricting others. To dynamically adjust penalty coefficients for different parameter groups, we present constrained parameter regularization (CPR) as an alternative to traditional weight decay. Instead of applying a single constant penalty to all parameters, we enforce an upper bound on a statistical measure (e.g., the $L_2$-norm) of parameter groups. Consequently, learning becomes a constraint optimization problem, which we address by an adaptation of the augmented Lagrangian method. CPR only requires two hyperparameters and incurs no measurable runtime overhead. Additionally, we propose a simple but efficient mechanism to adapt the upper bounds during the optimization. We provide empirical evidence of CPR's efficacy in experiments on the "grokking" phenomenon, computer vision, and language modeling tasks. Our results demonstrate that CPR counteracts the effects of grokking and consistently matches or outperforms traditional weight decay.

## 1 INTRODUCTION

Deep neural networks are the bedrock of many state-of-the-art machine learning applications (Schmidhuber, 2015). While these models have exhibited unparalleled expressivity, they also possess millions, sometimes trillions, of parameters (Fedus et al., 2022). This massive capacity makes them susceptible to overfitting, where models memorize nuances of the training data but underperform on unseen examples. To mitigate this, many different regularization techniques have been adopted, with weight decay and $L_2$ regularization being the most popular (Hanson & Pratt, 1988; Krogh & Hertz, 1991; Bos & Chug, 1996). $L_2$ regularization penalizes the squared magnitude of model parameters and (decoupled) weight decay (which is equivalent to $L_2$ regularization for non-adaptive gradient algorithms (Loshchilov & Hutter, 2019)) multiplies all weights with a constant at every step. This seemingly simple act offers numerous benefits by curbing the growth of individual weights, reducing the risk of relying on any particular feature excessively, and thus promoting model generalization.

However, not all parameters in a neural network have the same role or importance and different weights could benefit from different regularizations. Similarly, it is unclear if a single weight decay value is optimal for the entire duration of optimization, especially for large-scale training. Indeed, Ishii & Sato (2018) showed that a small deep learning model could benefit from layer-wise weight decay values, and various works showed that scheduling weight decay could improve final performance (Lewkowycz & Gur-Ari, 2020; Yun et al., 2020; Caron et al., 2021; Oquab et al., 2023). This indicates that a dynamic penalty for each individual parameter group could be beneficial for neural network training. Since scheduled or parameter-wise weight decay comes with additional hyperparameters (which are often sensitive to the task), we propose a different approach to obtain customized, dynamic parameter regularization. Instead of uniformly penalizing weights, we propose to keep them in a certain range, thus ensuring stability without imposing regularization where it is unnecessary. Constraining parameters, especially based on statistical measures like the $L_2$ norm, provide a flexible and adaptive form of regularization that accounts for the heterogeneity of parameters. In this paper, we propose *constrained parameter regularization (CPR)*, which enforces an upper bound on a statistical measure of individual parameter groups (e.g., a weight matrix in a linear layer). This allows us to reformulate regularization as a constrained optimization problem, which we address by an adaptation of the augmented Lagrangian method. The resulting regularization strength is individual to

each parameter group and adaptive over time while being configurable by only two hyperparameters. In the following, after discussing related work (Section 2) and background on weight decay and the augmented Lagrangian method (Section 3), we make the following contributions:

- We introduce CPR for individualized and dynamic weight regularization (Section 4). Specifically,
  - To avoid the need for separate penalty scaling coefficients, we formulate regularization as a constraint problem and derive CPR as the solution of this problem (Section 4.1).
  - We identify three different ways for initializing the value of this constraint (an upper bound on a statistical measure, such as $L_2$ norm or standard deviation; Section 4.2).
  - We propose a modification to automatically lower the bound during training (Section 4.3).
- We provide empirical evidence of the benefit of CPR by showing improved performance over weight decay in experiments on the "grokking" phenomenon (Section 5.1), computer vision (image classification with ResNets in Section 5.2 and medical image segmentation with a UNet in Section 5.4) and a language modeling task with mid-scale GPT2 (Section 5.3).
- We provide an open-source implementation of CPR which can be easily adapted by replacing the optimizer class.

## 2 RELATED WORK

Weight decay is an effective regularization technique to improve the generalization and model performance (Zhang et al., 2018), and the idea of adapting parameter regularization during the training is not new. Lewkowycz & Gur-Ari (2020) investigated the effect of $L_2$ regularization on overparameterized networks and found the time it takes the network to reach peak performance is proportional to the $L_2$ regularization parameter. They proposed an initialization scheme for $L_2$ regularization and an annealing schedule for the $L_2$ parameter. Yun et al. (2020) use a combination of weight decay scheduling and knowledge distillation to improve performance on computer vision tasks. More recent works on self-supervised vision transformers also use a weight decay schedule (Caron et al., 2021; Oquab et al., 2023). In contrast to our work, none of these proposes a dynamic and individual adaptation of each regularized parameter group. Also, a schedule comes with varying hyperparameter choices while CPR adapts arbitrarily many parameter groups with only two hyperparameters (out of which one is fixed in all our experiments). Instead of using a schedule, Nakamura & Hong (2019) proposes *AdaDecay*, where the $L_2$ penalty is scaled by standardized gradient norms and a sigmoid function. Another way to regularize parameters is to fix the norm of individual parameter groups (Salimans & Kingma, 2016) or to limit the total norm of all parameters (Liu et al., 2023) to a fixed value. This fixed value is a more sensitive hyperparameter than the hyperparameter in our work. Our proposed method is not the first to use Lagrangian methods in machine learning (Platt & Barr, 1987). Its application in deep learning so far focuses on variational methods and generative models: Rezende & Viola (2018) introduced the *Generalized ELBO with Constrained Optimization* algorithm to optimize VAEs using Lagrange multipliers optimized by the min-max scheme, and Kohl et al. (2018) and Franke et al. (2022) adapted the Lagrangian method from Rezende & Viola (2018) to train probabilistic U-nets and probabilistic Transformer models. While these works leverage the Lagrangian to handle several losses in joint optimization problems, our work leverages it to enable individual regularization strengths.

## 3 BACKGROUND

### 3.1 DISTINCTION BETWEEN WEIGHT DECAY AND $L_2$ REGULARIZATION

Regularization methods, such as $L_2$-regularization or weight decay are commonly used to restrict parameter updates and enhance generalization by reducing unnecessary complexity (Hanson & Pratt, 1988; Krogh & Hertz, 1991; Bos & Chug, 1996). Both can be motivated by introducing a "cost" to weight values. Specifically, in $L_2$-regularization, instead of minimizing only the loss function $L(\boldsymbol{\theta}, \boldsymbol{X}, \boldsymbol{y})$ with parameters $\boldsymbol{\theta}$ and data $\mathcal{D} = \{(\boldsymbol{X}_n, \boldsymbol{y}_n)\}_{n=0}^{N}$, a weighted penalty (regularization) term $R(\boldsymbol{\theta})$ is added to the loss, resulting in the training objective

$$\min_{\boldsymbol{\theta}} \quad L(\boldsymbol{\theta}, \boldsymbol{X}, \boldsymbol{y}) + \gamma \cdot R(\boldsymbol{\theta}),$$

where $R(\boldsymbol{\theta}) = \frac{1}{2}\|\boldsymbol{\theta}\|_2^2$ denotes the regularization function and $\gamma \in \mathbb{R}^+$ the strength of the penalty. On the other hand, weight decay directly modifies the update rule of the parameters, such that

$$\boldsymbol{\theta}_{t+1} \leftarrow \boldsymbol{\theta}_t + \mathrm{Opt}(L, \eta) - \eta \cdot \gamma \cdot \boldsymbol{\theta}_t,$$

where $\mathrm{Opt}(L, \eta)$ denotes an optimizer providing the gradient-based update at iteration $t$ and $L = L(\boldsymbol{\theta}_t, \boldsymbol{X}_t, \boldsymbol{y}_t)$ the loss. For example, $\mathrm{Opt}(L, \eta) = -\eta \cdot \nabla_{\boldsymbol{\theta}} L(\boldsymbol{\theta}_t, \boldsymbol{X}_t, \boldsymbol{y}_t)$ with learning rate $\eta \in \mathbb{R}^+$ in case of gradient descent. Thus, the main difference between weight decay and $L_2$-regularization is that the gradients of the regularization accumulate in momentum terms in the case of $L_2$-regularisation, while they are treated separately in (decoupled) weight decay. This has also been extensively discussed by Loshchilov & Hutter (2019) with the introduction of the AdamW optimizer.

## 3.2 THE AUGMENTED LAGRANGIAN METHOD

We briefly review the augmented Lagrangian method, see e.g. Bertsekas (1996), which our method is based on. For the derivation, we follow the motivation of Nocedal & Wright (2006, pp. 523-524).

Consider the following inequality-constrained optimization problem

$$\underset{\boldsymbol{x}}{\mathrm{minimize}} \; f(\boldsymbol{x}) \quad \text{s.t.} \quad c(\boldsymbol{x}) \le 0,$$

with $f(\boldsymbol{x}) : \mathbb{R}^n \to \mathbb{R}$ and a constraint $c(\boldsymbol{x}) : \mathbb{R}^n \to \mathbb{R}$. One way to address the constraint is to find an equivalent, unconstrained problem with the same optimal solution. For example,

$$\underset{\boldsymbol{x}}{\mathrm{minimize}} \; F(\boldsymbol{x}) \quad \text{with} \quad F(\boldsymbol{x}) = \max_{\lambda \ge 0} \; f(\boldsymbol{x}) + \lambda \cdot c(\boldsymbol{x}). \tag{1}$$

Evidently, for any infeasible point $\boldsymbol{x}$ with $c(\boldsymbol{x}) > 0$, $\lambda \cdot c(\boldsymbol{x})$ in the inner maximization can yield arbitrarily high values ($\to \infty$). Thus, any solution candidate must clearly be feasible. Unfortunately, $F(\boldsymbol{x})$ is not suitable for gradient-based optimization, as it provides no useful gradient information to restore feasibility. To alleviate this problem, we consider a smooth approximation of $F(\boldsymbol{x})$, namely

$$\hat{F}(\boldsymbol{x}, \bar{\lambda}, \mu) = \max_{\lambda \ge 0} \; f(\boldsymbol{x}) + \lambda \cdot c(\boldsymbol{x}) - \frac{1}{2\mu}(\lambda - \bar{\lambda})^2, \tag{2}$$

where $\bar{\lambda} \in \mathbb{R}$ may be seen as a point we wish to remain proximal to and $\mu \in \mathbb{R}^+$ as a factor determining the strength with which this proximity is enforced. For $\mu \to \infty$, $\hat{F}(\boldsymbol{x}, \bar{\lambda}, \mu) \to F(\boldsymbol{x})$.

The maximization in $\hat{F}(\boldsymbol{x}, \bar{\lambda}, \mu)$ has a closed form solution with $\lambda^\star = (\bar{\lambda} + \mu \cdot c(\boldsymbol{x}))^+$, where $(\cdot)^+ = \max\{0, \cdot\}$, see Appendix A for the derivation.

Consequently,

$$\hat{F}(\boldsymbol{x}, \bar{\lambda}, \mu) = f(\boldsymbol{x}) + h(\boldsymbol{x}, \bar{\lambda}, \mu) \tag{3}$$

with

$$h(\boldsymbol{x}, \bar{\lambda}, \mu) = \begin{cases} c(\boldsymbol{x})(\bar{\lambda} + \frac{\mu}{2}c(\boldsymbol{x})), & \text{if} \quad \bar{\lambda} + \mu \cdot c(\boldsymbol{x}) \ge 0 \\ -\frac{1}{2\mu}\bar{\lambda}^2 & \text{else.} \end{cases} \tag{4}$$

The constraint thus only interferes with the minimization (gradient) of $f(\boldsymbol{x})$ if $(\bar{\lambda} + \mu \cdot c(\boldsymbol{x}))^+ \ge 0$.

We can now try to solve the unconstrained problem $\hat{F}(\boldsymbol{x}, \bar{\lambda}, \mu)$ with familiar methods, such as gradient descent, and obtain an approximate solution to the original problem. Specifically, the gradient of $\hat{F}(\boldsymbol{x}, \bar{\lambda}, \mu)$ with respect to $\boldsymbol{x}$ is given by

$$\nabla_{\boldsymbol{x}} \hat{F}(\boldsymbol{x}, \bar{\lambda}, \mu) = \nabla_{\boldsymbol{x}} f(\boldsymbol{x}) + \lambda^\star \cdot \nabla_{\boldsymbol{x}} c(\boldsymbol{x}). \tag{5}$$

The quality of the approximation, and thus the solution, clearly depends on $\mu$ and $\bar{\lambda}$. However, after solving $\hat{F}(\boldsymbol{x}, \bar{\lambda}, \mu)$ for some value of $\bar{\lambda}$, we can perform an update step $\bar{\lambda} \leftarrow \lambda^\star$ and attempt to perform minimization again. Intuitively, if the previous minimization of $\hat{F}(\boldsymbol{x}, \bar{\lambda}, \mu)$ resulted in an infeasible solution with $c(\boldsymbol{x}) > 0$, $\bar{\lambda}$ is increased. Hence, the next minimization of $\hat{F}(\boldsymbol{x}, \bar{\lambda}, \mu)$ likely results in a solution with less constraint violation. On the other hand, if $c(\boldsymbol{x}) \le 0$, $\bar{\lambda}$ is decreased. Subsequently, the influence of the constraint is decreased. This loop of alternating minimization of $\hat{F}(\boldsymbol{x}, \bar{\lambda}, \mu)$ and update to $\bar{\lambda}$ can be repeated until a sufficiently good solution is found or the procedure converges if $\bar{\lambda}$ does not receive updates anymore.

For multiple constraints $c_j(\boldsymbol{x})$, $j = 1, \cdots, J$, the above can be readily extended with a multiplier $\lambda^j$ for each constraint. Since the maximization in the smooth approximation is separable in the $\lambda^j$, the same update rule may be applied for each $\lambda^j$ separately using on the respective constraint $c_j(\boldsymbol{x})$.

# 4 CONSTRAINED PARAMETER REGULARIZATION

In this section, we introduce Constrained Parameter Regularization (CPR), where we adapt the augmented Lagrangian method to enforce upper bounds on regularization terms. Compared to classical regularization, with a fixed regularization coefficient $\gamma$, the proposed approach will allow for variable regularization coefficients $\lambda^j$ (Lagrange multipliers) for $j = 1, \cdots, J$ parameter groups $\boldsymbol{\theta}^j \subseteq \boldsymbol{\theta}$ that should be regularized. These regularization coefficients are updated alongside the network parameters $\boldsymbol{\theta}$.

## 4.1 REGULARIZATION THROUGH CONSTRAINTS

Classical weight decay, as introduced earlier, is used as a means to restrict the freedom of parameter adaptation. This restriction is applied with a scaling factor $\gamma$ (hyperparameter) and applies uniformly to all parameters. However, we conjecture that applying an individual adaptation pressure instead may be beneficial. Unfortunately, this would require a separate coefficient for each parameter group where a separate weight decay should be applied. To avoid the need for separate scaling coefficients, we formulate regularization as a constrained problem. Here, the loss function $L(\boldsymbol{\theta}, \boldsymbol{X}, \boldsymbol{y})$, with network parameters $\boldsymbol{\theta}$, takes the place of the objective. Consequently, the learning problem becomes

$$\underset{\boldsymbol{\theta}}{\text{minimize }} L(\boldsymbol{\theta}, \boldsymbol{X}, \boldsymbol{y}) \quad \text{s.t.} \quad c_j(\boldsymbol{\theta}^j) = R(\boldsymbol{\theta}^j) - \kappa^j \leq 0, \quad \text{for} \quad j = 1, \cdots, J, \quad (6)$$

where $R(\boldsymbol{\theta}^j)$ is a regularization function (e.g., the $L_2$-norm in case of weight decay) for a parameter group $\boldsymbol{\theta}^j \subseteq \boldsymbol{\theta}, j = 1, \cdots, J$, and $\kappa \in \mathbb{R}$ denotes a chosen bound.

To solve Equation 6, we follow the augmented Lagrangian method with slight modifications. First, instead of performing a full optimization of the loss before updating $\bar{\lambda}$, we perform updates in every step. This is motivated by the fact that full optimization is generally infeasible in a deep learning setting. Moreover, similar to the difference between weight decay and $L_2$-regularization, we treat the update between the loss-dependent and the constraint-dependent part separately. Hence, instead of introducing $\hat{L}(\boldsymbol{x}, \bar{\lambda}, \mu)$ analogously to Equation 2, and performing optimization on this objective, we independently apply updates for both steps. Consequently, the constraint violations do not accumulate in momentum terms. We also remove the influence of the learning rate on the regularization. From a practical perspective, our modification does not interfere with gradient-based optimization algorithms and can be readily combined with any such optimizer. The full algorithm is given by Algorithm 1.

---

**Algorithm 1** Optimization with constrained parameter regularization (CPR) .

**Require:** Loss Function $L(\boldsymbol{\theta}, \boldsymbol{X}, \boldsymbol{y})$ with parameters $\boldsymbol{\theta}$, and data $\mathcal{D} = \{(\boldsymbol{X}_n, \boldsymbol{y}_n)\}_{n=0}^N$
**Require:** Hyperparameters: Learning rate $\eta \in \mathbb{R}^+$, Lagrange multiplier update rate $\mu \in \mathbb{R}^+$
**Require:** Optimizer $\mathrm{Opt}(\cdot)$ for minimization, Regularization function $R(\boldsymbol{\theta})$ (e.g. L2-norm)
1: # Initialization
2: $t \leftarrow 0$
3: $\boldsymbol{\theta}_t \leftarrow \mathrm{Initialize}(L(\cdot))$
4: $\lambda_t^j \leftarrow 0$ for $j = 1, \cdots, J$
5: $\kappa^j \leftarrow \mathrm{Initialize}(\boldsymbol{\theta}_0^j)$ for $j = 1, \cdots, J$         ▷ Initializing the upper bound, see Section 4.2
6: # Training
7: **for** $\boldsymbol{X}_t, \boldsymbol{y}_t \sim \mathcal{D}$ **do**
8:     $\boldsymbol{\theta}_{t+1} \leftarrow \boldsymbol{\theta}_t + \mathrm{Opt}(L(\boldsymbol{\theta}_t, \boldsymbol{X}_t, \boldsymbol{y}_t), \eta)$       ▷ Classic parameter update using, e.g., Adam.
9:     **for** each regularized parameter group $\boldsymbol{\theta}_t^j$ in $\boldsymbol{\theta}_t$ **do**
10:         $\lambda_{t+1}^j \leftarrow \left(\lambda_t^j + \mu \cdot (R(\boldsymbol{\theta}_t^j) - \kappa^j)\right)^+$
11:         $\boldsymbol{\theta}_{t+1}^j \leftarrow \boldsymbol{\theta}_{t+1}^j - \nabla_{\boldsymbol{\theta}^j} R(\boldsymbol{\theta}_t^j) \cdot \lambda_{t+1}^j$
12:     **end for**
13:     $t \leftarrow t + 1$
14: **end for**

---

Conceptually, the method can be understood as the $\lambda^j$ accumulating constraint function values (weighted with $\mu$) over the iterations. These then increase (or decrease) the influence of the constraint (via its gradient) on the search direction. When points in the feasible domain are found for which $c_j(\boldsymbol{\theta}) \leq 0$, $\lambda$ is decreased until it eventually reaches 0. If, on the other hand, the optimal solution lies on the boundary, where $c_j(\boldsymbol{\theta}) = 0$, $\lambda$ should converge to a value $\lambda^\star$ where the update direction of the optimizer and the gradient of the constraints cancel each other. However, this situation is unlikely to occur in a deep learning setting due to the stochasticity of minibatches and potential adaptations to the learning rate.

## 4.2 INITIALIZATION OF BOUNDS

The upper bound $\kappa$ is the most crucial hyperparameter for CPR, and we identify three ways to initialize it. (1) Set $\kappa$ uniform (`Kappa-K`): Set one value for all regularized parameter groups as an initial value for the upper bound, $\kappa \in \mathbb{R}^+$. (2) Set $\kappa$ based on $\theta$-initialization (`Kappa-kI`$_0$): Initialize the upper bound based on the initial parameter groups' regularization function, which could be affected by a parameter group's individual size and/or initialization scheme (e.g. a depth-dependent initialization): $\kappa^i = k \cdot R(\boldsymbol{\theta}_{t=0}^i)$, with $k \in \mathbb{R}^+$ as the factor of the initial measure. (3) Set $\kappa$ with warm start (`Kappa-I`$_s$): Instead of selecting a factor $k$ of the initial regularization function, train our model parameters $\boldsymbol{\theta}$ for a specific number of update steps and then bind the regularization to the current regularization function value: $\kappa^i = R(\boldsymbol{\theta}_{t=s}^i)$, with $s \in \mathbb{N}^+$ as a hyperparameter for the start of the regularization; please find an integration in CPR in Appendix B. In terms of search space for the optimal initialization, the `Kappa-I`$_s$ initialization is practically the simplest since it can only be a natural number between zero and the maximal training steps. All three initialization variations have in common that they require only one hyperparameter despite the fact that `Kappa-kI`$_0$ and `Kappa-I`$_s$ initialize each parameter group independently.

## 4.3 ADAPTIVE BOUNDS

While the Lagrangian multiplier introduces individual regularization pressure to each parameter group $\boldsymbol{\theta}^j$, it does so only in case of recent constraint violations (if $\lambda^j > 0$). If the bound on the regularization for a parameter group was set too high, the parameter group may not be exposed to any regularization pressure over the course of training. This contrasts with weight decay, where continuous pressure is applied to enhance generalization throughout the training. To emulate the continuous pressure of weight decay, we propose an adaptive mechanism to adjust the upper regularization bound during training. This can be achieved by leveraging existing states. Specifically, the value of $\lambda^j$ offers insights into constraint violations. When $\lambda^j = 0$, the constraint $c_j(\boldsymbol{\theta})$ can be regarded as inactive. In this case, we may consider adjusting its bound $\kappa^j$ to align with the current constraint value of $c(\boldsymbol{\theta}_j)$. To implement these adaptive bounds, we add a conditional update rule for $\kappa^j$ after our CPR update. It updates the upper bound for each parameter group $\theta_t^j$ individually by

$$\kappa_{t+1}^j \leftarrow \begin{cases} R(\theta_t^j) & \text{if } \lambda_t^j = 0 \text{ and } \lambda_{t-1}^j > 0 \\ \kappa_t^j & \text{otherwise,} \end{cases}$$

where $\lambda_{t-1}^j > 0$ indicates that the upper bound was previously violated and $c_j(\boldsymbol{\theta}^j)$ was active. Consequently, this enables a gradual reduction of the bounds $\kappa^j$ over the course of training without exerting excessive pressure on the optimization process. We dub this extension *AdaCPR* and the complete algorithm can be seen in Appendix C.

## 5 EXPERIMENTS

We now describe various experiments to understand CPR and its upper bound better. Preliminary experiments showed that $\mu$ is not a sensitive hyperparameter and we chose $\mu = 1.0$ for all our experiments (see experiments in Appendix D). This leads to the upper bound $\kappa$ as the sole hyperparameter of CPR. We also aim to emphasize one of the initialization methods of $\kappa$ for general usage, evaluate different regularization functions, and provide empirical evidence for the practical usefulness of CPR. In this work, we consider a weight matrix in a neural network as an individual parameter group and regularize all parameters in a network except for biases and normalization weights.

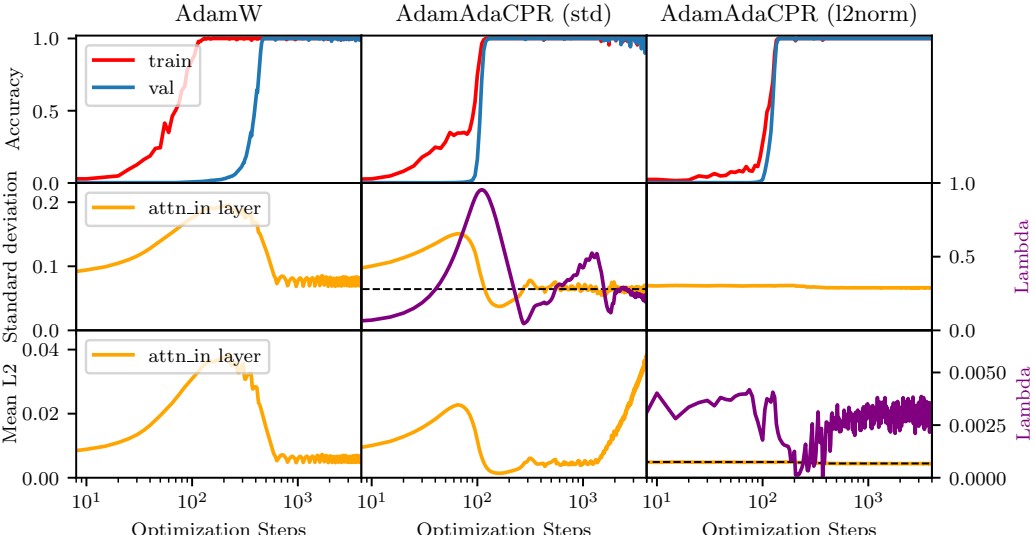

Figure 1: Experiments on the modular addition task to illustrate the effect of AdaCPR. The training steps on the x-axis are displayed in log scale. The training and validation accuracy are displayed in red and blue. In the middle row, we see the standard deviation of the *attention-in* weight parameter during the training progress and at the bottom the mean $L_2$ norm. In the left column, we use AdamW for optimization with a weight decay of $1.0$, and in the middle and right columns Adam with AdaCPR. On the left side, we see the $\lambda$ value during training and the dotted black line represents the upper bound $\kappa$. We optimize in the middle to a standard deviation of $k = 0.9$ times the initial value and on the right to an $L_2$ norm with $k = 0.8$.

## 5.1 MODULAR ADDITION

In the realm of neural network regularization, the phenomenon of *grokking* has garnered significant attention. As discovered by Power et al. (2021), grokking is characterized by a sudden generalization after prolonged training without discernible improvements in training or validation loss. We train a 1-layer Transformer on the modular addition task which is the primary benchmark for studying this phenomenon. To explore the possibilities of CPR we consider two regularization variations: one constraint based on the $L_2$ norm and one on the standard deviation. The standard deviation is interesting since it does not constrain the weight parameters to be centered around zero. We use `Kappa-kI0` for the initialization of $\kappa$ with a factor of $k = 0.8$ for the $L_2$ norm and $k = 0.9$ for the standard deviation. We found these factors by a small grid search influenced by the rescaling described by Liu et al. (2023). A comprehensive list of all hyperparameters can be found in Appendix E.

We now compare AdamW to Adam with CPR (*AdamCPR*) and Adam with AdaCPR (*AdamAdaCPR*). The results in Figure 1 reveal that AdaCPR nearly mitigates the effects of grokking and achieves faster convergence. Both constraint variations, the standard deviation, and $L_2$ norm successfully bridge the performance gap between training and validation by dynamically regularizing parameters. Notably, the CPR based on standard deviation exhibited a more uniform behavior across the weight matrix. But at the end of the training, the $L_2$ norm starts to increase which could indicate an unstable behavior in the long run and could be caused by not encouraging a zero-centered parameter distribution. In contrast, the $L_2$-constrained training demonstrated long-term stability. A unique feature of our approach is the individual adaptation of each parameter. For a granular analysis of each layer's behavior, we point to additional plots in Appendix E, where we see individual $\lambda$ adaptions over the training progress.

Recent studies explored various strategies to counteract the effects of grokking (Pearce et al., 2023). Notably, the approach of rescaling parameters to an initial weight norm has been highlighted as a potential solution (Liu et al., 2023). While this method does offer some mitigation against grokking, our observations indicate that such training tends to be more unstable. We compare rescaling to CPR in Figure E.1. We also see that CPR displays a slightly more unstable behavior than AdaCPR.

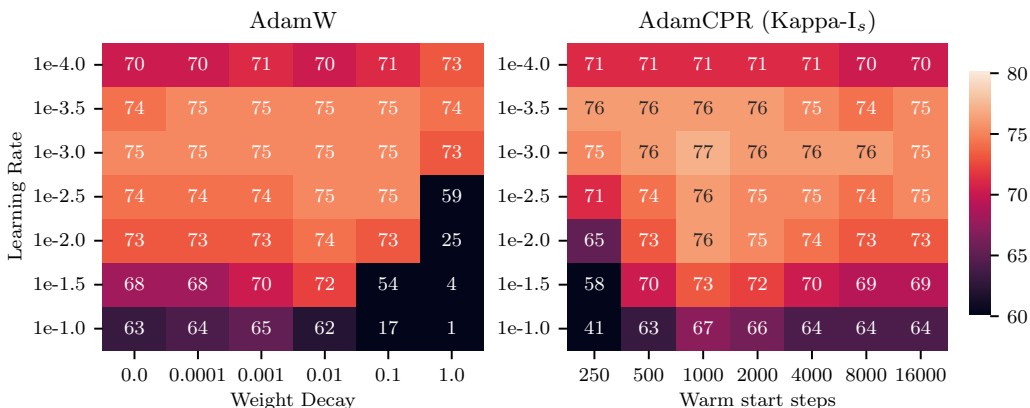

Figure 2: Percentage of correct labels of the ResNet18 trained on the CIFAR100 with AdamW (left) and AdamCPR (right). AdamCPR uses the $L_2$ norm as a regularization function and `Kappa-I`$_s$.

This could be caused by the constant pressure on the regularization due to the adaptation. It should be mentioned that when we increase the weight decay regularization in AdamW, we can also close the gap between training and validation performance. Unfortunately, this comes at the price of unstable training behavior. We refer the interested reader to Figure E.6 and Figure E.7. To illustrate the performance for different bounds, we provide plots of single runs with CPR and AdaCPR on the regularization with standard deviation and $L_2$ norm in Figures E.2, E.3, E.4, and E.5. We see that adaptation stabilized the training since we corrected the bound and kept the pressure on the optimization goal. We trained the grokking experiments within minutes per run on a consumer GPU without a runtime difference between AdamW and AdamCPR. Since we observed a more unstable training behavior when using the standard deviation as a regularization function, we resort to the $L_2$ norm in the following experiments.

## 5.2 IMAGE CLASSIFICATION

To evaluate CPR's effectiveness and design choices, we tested CPR in image classification using ResNet18 on the CIFAR100 dataset (He et al., 2016; Krizhevsky, 2009). We compared AdamW to AdamCPR and AdamAdaCPR with the three initializations Section 4.2 and $L_2$-norm as a measure for the regularization constraint. For the $\kappa$ initialization `Kappa-K`, we use a range of $\kappa = [0.005, \ldots, 0.16]$, for `Kappa-kI`$_0$ a range of $k = [4, \ldots, 256]$, and for `Kappa-I`$_s$ a range of $s = [250, \ldots, 4000]$ steps. Thus, the warmup steps we used for $\kappa$ are in the same range as the learning rate warmup (500 steps). The ResNet18 was trained on a consumer GPU in 15-20 minutes. There was no significant runtime difference between AdamW and AdamCPR. For a detailed list of training hyperparameters, we refer the reader to Table F.1. Figures 2 show the best mean validation performance for different learning rates and regularization hyperparameters of the best-performing initialization. Please find plots for all initialization methods in Figure F.1.

Further, we compared our method to related work. We used Adam with rescaling of parameters from Liu et al. (2023) and show the results in Figure F.2. We evaluated a cosine schedule for a decreasing and increasing the weight decay value similar to Caron et al. (2021); Oquab et al. (2023). The results can be found in Figure F.3. It should be mentioned that Yun et al. (2020) also performed weight decay scheduling on CIFAR100 with the use of a ResNet18. Since their code was not published, we point to Figure 3 of their experimental results, where an accuracy of around $60\%$ was reported which is below our AdamW baseline. Furthermore, we implemented AdaDecay Nakamura & Hong (2019) and evaluated the method for different alpha values, see Figure F.4.

While the related works we compared to do outperform AdamW in some configurations, they do not perform better than AdamCPR with `Kappa-I`$_s$. Furthermore, we found that initializing with `Kappa-kI`$_0$ performs better than selecting a uniform $\kappa$ in `Kappa-K`. This may be explained by the value of the regularization function depending on the size of the jointly regularized parameter group and initialization method. The warm start $\kappa$ initialization method, `Kappa-I`$_s$, performed the best. The reason for this may lie in its general flexibility, as, in a sense, warm started bounds may

Table 1: The mean results of the GPT2s training over 200k steps. The values below the method denote the weight decay factor $\gamma$ in case we use AdamW. For CPR and AdaCPR, they indicate the number of warm-start steps $s$ of the initialization `Kappa-I`$_s$. The $L_2$ norm is used as a regularization function. Please find corresponding standard deviations in Table G.1

| GPT2s | AdamW | | | CPR | | | AdaCPR | | |
|---|---|---|---|---|---|---|---|---|---|
| 200k | 1e-3 | 1e-2 | 1e-1 | 5k | 10k | 20k | 5k | 10k | 20k |
| Accuracy ↑ | 0.445 | 0.446 | 0.441 | 0.445 | 0.447 | 0.446 | 0.445 | 0.447 | 0.446 |
| PPL ↓ | 17.98 | 17.84 | 18.58 | 17.96 | 17.68 | 17.80 | 17.95 | 17.69 | 17.79 |

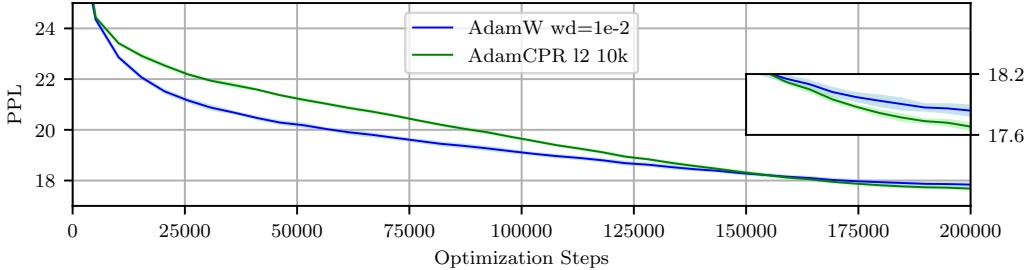

Figure 3: Experiments on OpenWebText and a GPT2s model. The mean validation PPL of three runs (±std as shaded area) with AdamW and the CPR (`Kappa-I`$_s$) are displayed in blue and green respectively. We can notice that CPR regularized the training more at the beginning of the training.

be considered "learned", since they reflect the actual magnitudes and distributions of the parameter groups in the training process. This can subsequently constrain the network to find a suitable solution in a region of the parameter space that displayed healthy learning behavior before. Finally, we want to note the seemingly linear dependence between the learning rate and well-performing initialization factors $k$ when using `Kappa-kI`$_0$ in Figure F.1. This may suggest the possibility of deriving initialization rules based on the intended learning rate and initial parameter values in future works.

## 5.3 LANGUAGE MODELLING

We also performed experiments training a GPT2s language model (Radford et al., 2019) on Open-webtext (Gokaslan & Cohen, 2019) with three random seeds. For an efficient implementation, we use flash attention (Dao et al., 2022) and rotary position embedding (Su et al., 2021). In this experiment, we compared AdamW on different weight decay values to AdamCPR and AdamAdaCPR. We use the $L_2$-norm as a measure for the regularization constraint and the warm-started $\kappa$ initialization `Kappa-I`$_s$ as this performed best in previous experiments. We use a learning rate warmup for 5k steps, a cosine annealing, and train for 200k steps. We orientate the warmup steps of $\kappa$ based on the warmup steps of the learning rate and evaluated for initializing $\kappa$ after 5k, 10k, and 20k steps. The complete hyperparameters can be found in Appendix G. The GPT2s model is trained on 8 A100 GPUs and the runtime for AdamW was between 12.6h and 12.8h, and for AdamCPR between 12.4h and 12.9h. This indicates no additional runtime overhead for our method on larger-scale models. The mean results in Table 1 suggest that AdamCPR and AdamAdaCPR outperform the best weight decay configuration and that the selection for the warmup time for AdamCPR seems to not be very sensitive in this case, please find the standard deviation of the results in Appendix G. We show the learning curves of the best AdamW and best AdamCPR run in Figure 3. We observe that weight decay regularizes the model less strongly in the early stages of training which may lead to better final performance. To investigate the scalability of our results above, we also performed experiments over a longer runtime with twice the optimization steps (GPT2s 400k) and on a larger model with the GPT2m settings (dim. 1024 / 24 layers / 354M parameters instead of 768/12/124M). We used the same weight decay and `Kappa-I`$_s$ initialization as in the GPT2s experiment. We find again that AdamCPR outperforms AdamW, which could indicate that AdamCPR is also capable of optimizing larger models or longer training. Please find the results in Table 2.

Table 2: To demonstrate the scalability of our approach we performed with the best settings from the GPT2s 200k experiment runs over a longer runtime (GPT2s 400k) and on a larger model (GPT2m).

| GPT2s 400k | AdamW 1e-2 | AdamCPR 10k |
|---|---|---|
| Accuracy ↑ | 0.449 | **0.450** |
| PPL ↓ | 17.43 | **17.32** |

| GPT2m 200k | AdamW 1e-2 | AdamCPR 10k |
|---|---|---|
| Accuracy ↑ | 0.472 | **0.474** |
| PPL ↓ | 14.23 | **14.03** |

## 5.4 MEDICAL IMAGE SEGMENTATION

To demonstrate the general effectiveness of the proposed CPR approach, we also evaluate it in the context of medical image segmentation. We test the proposed method on three segmentation benchmarks, the Multi-Atlas Labeling Beyond the Cranial Vault (BTCV) Landman et al. (2015) task, the Heart Segmentation task of the Medical Segmentation Decathlon Antonelli et al. (2022) and the 2020 version of the Brain Tumor Segmentation challenge (BraTS) task Menze et al. (2015). Here, we make use of the data pipeline and network architectures following the nnU-Net framework (Isensee et al., 2021), which is regarded as the state-of-the-art framework for medical image segmentation. We implement a training schedule with a total of 25k steps (for the Heart and BraTS tasks) and 125k steps for BTCV. We introduce a learning rate warmup for the first 2k steps, after which the learning rate is annealed following a polynomial schedule, see hyperparameters in Appendix H. We present the results in Table 3, where different weight decay configurations in AdamW are evaluated to AdamCPR with the $L_2$ norm and $\texttt{Kappa-I}_s$ initialization. We report the commonly used Dice scores, averaged across cross-validation folds. These results indicate that CPR surpasses even the best AdamW results. However, we note that applying $\texttt{Kappa-I}_s$ initialization too late can cause instabilities during the training of the typical U-Net architectures seen in medical imaging due to weak regularization.

Table 3: Results of medical image segmentation training on the BTCV, Heart, and BraTS datasets. We show the mean Dice score across 5 folds (3 for BTCV) for a range of weight decay values ($\gamma$) for AdamW and AdamCPR for different warmup steps $s$.

| | AdamW | | | | | AdamCPR | | | |
|---|---|---|---|---|---|---|---|---|---|
| | 1e-5 | 1e-4 | 1e-3 | 1e-2 | 1e-1 | 1k | 2k | 3k | 4k |
| BTCV | 83.04 | 83.1 | 83.17 | 83.99 | 73.92 | 81.17 | 84.14 | **84.23** | 55.41 |
| Heart | 92.92 | 92.75 | 92.88 | 92.9 | 92.85 | 92.77 | **93.18** | 93.16 | 74.44 |
| BraTS | 75.85 | 76.01 | 76.22 | 76.12 | 75.42 | 75.29 | 76.46 | **76.65** | 75.63 |

## 6 CONCLUSION & FUTURE WORK

In this work, we introduce constrained parameter regularization (CPR), a method for regularization of neural network training via constraints. By enforcing an upper bound on a regularization function, we achieve effective regularization of the neural network training across various tasks. The constraints are handled by an adaptation of the augmented Lagrangian method without notable runtime overhead over standard weight decay. We provide empirical evidence for the capabilities of CPR when combined with Adam by improving over AdamW on modular addition, image classification, language modeling, and image segmentation tasks. From the perspective of the optimization loop, CPR can be combined with any gradient-based optimizer and requires only a minor addition to the training loop comparable to a learning rate scheduler. However, CPR still requires a hyperparameter $\kappa$ which needs to be tuned for a superior performance.

Future works could focus on a more efficient initialization of $\kappa$ and could e.g. investigate a connection between the learning rate, the parameter initialization, and the bound on the regularization function. These could provide insights for the development of automated $\kappa$ optimization schemes and may hold the potential to discover scaling rules to devise a stable default setting of $\kappa$ across problems. Further, CPR may be combined with different regularization functions and could be evaluated in long-run experiments to explore the effect of this kind of regularization in later stages of training.

## REPRODUCIBILITY STATEMENT

To ensure the reproducibility of our work, we included the source code in the supplementary materials, which we used to recreate the results. All experiments are implemented in Python with the use of PyTorch (Paszke et al., 2019). The source code contains scripts to replicate the grokking, image classification, language modeling, and medical image segmentation experiments. After installing the environment for which we added a setup script, one can run the grokking experiment within minutes on a consumer GPU and without any additional data. The image classification experiment and the language model experiment download the data at the first start. Please find detailed commands to run the single experiments in a *Readme* file.

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

# APPENDIX

## A  DERIVATION OF THE LAGRANGE MULTIPLIER UPDATE

For simplicity, we consider a single constraint. Note that multiple constraints can be addressed separately as the optimization problem would be separable in the respective $\lambda^j$. We need to solve

$$\underset{\lambda \geq 0}{\text{maximize}} \; f(\boldsymbol{x}) + \lambda \cdot c(\boldsymbol{x}) - \frac{1}{2\mu}(\lambda - \bar{\lambda})^2.$$

The optimal point of this problem is equivalent to the optimal point of

$$\underset{\lambda}{\text{minimize}} \; - f(\boldsymbol{x}) - \lambda \cdot c(\boldsymbol{x}) + \frac{1}{2\mu}(\lambda - \bar{\lambda})^2 \quad \text{s.t.} \quad -\lambda \leq 0.$$

To find candidates for optimal points, we need to solve the Karush–Kuhn–Tucker (KKT) system with the Lagrange function $\mathcal{L}(\lambda, \psi)$ and the Lagrange multiplier $\psi$

$$\mathcal{L}(\lambda, \psi) = -f(\boldsymbol{x}) - \lambda \cdot c(\boldsymbol{x}) + \frac{1}{2\mu}(\lambda - \bar{\lambda})^2 - \psi \cdot \lambda$$

Which leads to the KKT system

$$\nabla_\lambda \mathcal{L}(\lambda, \psi) = 0 \iff 0 = -c(\boldsymbol{x}) + \frac{1}{\mu}(\lambda - \bar{\lambda}) - \psi$$
$$\nabla_\psi \mathcal{L}(\lambda, \psi) \leq 0 \iff 0 \geq -\lambda$$
$$\lambda \cdot \psi = 0 \tag{7}$$

According to the complementary conditions Equation 7, the constraint is either active, hence $\lambda = 0$ and $\psi \geq 0$ or inactive, such that $\lambda > 0$, and consequently, $\psi = 0$.

**Case**: $\lambda = 0$ and $\psi \geq 0$

Here, $\lambda = 0$ (by assumption), and $\psi$ is given by

$$\nabla_\lambda \mathcal{L}(\lambda, \psi) = 0 \iff 0 = -c(\boldsymbol{x}) + \frac{1}{\mu}(0 - \bar{\lambda}) - \psi$$
$$\psi = -c(\boldsymbol{x}) - \frac{\bar{\lambda}}{\mu}$$

Since we require $\psi \geq 0$ for a KKT point, (note that $\mu > 0$)

$$0 \leq \psi = -c(\boldsymbol{x}) - \frac{\bar{\lambda}}{\mu}$$
$$\iff 0 \leq -\mu \cdot c(\boldsymbol{x}) - \bar{\lambda}$$
$$\iff 0 \geq \bar{\lambda} + \mu \cdot c(\boldsymbol{x})$$

Consequently, $\lambda = 0$ is a candidate for the optimal point only when $0 \geq \bar{\lambda} + \mu \cdot c(\boldsymbol{x})$.

**Case**: $\lambda > 0$ and $\psi = 0$ (inactive constraint)

For this case we get

$$\nabla_\lambda \mathcal{L}(\lambda, \psi) = 0 = -c(\boldsymbol{x}) + \frac{1}{\mu}(\lambda - \bar{\lambda}) - 0$$
$$0 = -\mu \cdot c(\boldsymbol{x}) + \lambda - \bar{\lambda}$$
$$\lambda = \bar{\lambda} + \mu \cdot c(\boldsymbol{x})$$

Due to the geometry of the problem (quadratic with bound constraint), $\lambda = 0$ is the optimal solution if the constraint is active, i.e., if $\psi \geq 0$, which is the case if $0 \geq \bar{\lambda} + \mu \cdot c(\boldsymbol{x})$. Consequently, the optimal solution is given by

$$\lambda^\star = (\bar{\lambda} + \mu \cdot c(\boldsymbol{x}))^+. \tag{8}$$

Plugging this into $\hat{F}(\boldsymbol{x}, \bar{\lambda}, \mu)$, we get

$$\hat{F}(\boldsymbol{x}, \bar{\lambda}, \mu) = \begin{cases} f(\boldsymbol{x}) + c(\boldsymbol{x})(\bar{\lambda} + \frac{\mu}{2}c(\boldsymbol{x})), & \text{if} \quad \bar{\lambda} + \mu \cdot c(\boldsymbol{x}) \geq 0 \\ f(\boldsymbol{x}) - \frac{1}{2\mu}\bar{\lambda}^2, & \text{else} \end{cases}$$

And the gradient with respect to $\boldsymbol{x}$ is

$$\nabla_{\boldsymbol{x}}\hat{F}(\boldsymbol{x}, \bar{\lambda}, \mu) = \begin{cases} \nabla_{\boldsymbol{x}}f(\boldsymbol{x}) + \nabla_{\boldsymbol{x}}c(\boldsymbol{x})(\bar{\lambda} + \mu \cdot c(\boldsymbol{x})), & \text{if} \quad \bar{\lambda} + \mu \cdot c(\boldsymbol{x}) \geq 0 \\ \nabla_{\boldsymbol{x}}f(\boldsymbol{x}) - 0 & \text{else} \end{cases}$$

Or more compactly by using Equation 8

$$\nabla_{\boldsymbol{x}}\hat{F}(\boldsymbol{x}, \bar{\lambda}, \mu) = \nabla_{\boldsymbol{x}}f(\boldsymbol{x}) + \nabla_{\boldsymbol{x}}c(\boldsymbol{x}) \cdot \lambda^\star.$$

## B    THE CPR ALGORITHM WITH KAPPA-I$_s$

---

**Algorithm 2** Optimization with constrained parameter regularization (CPR) and Kappa-I$_s$ .

---

**Require:** Loss Function $L(\boldsymbol{\theta}, \boldsymbol{X}, \boldsymbol{y})$ with parameters $\boldsymbol{\theta}$, and data $\mathcal{D} = \{(\boldsymbol{X}_n, \boldsymbol{y}_n)\}_{n=0}^{N}$
**Require:** Hyperparameters: Learning rate $\eta \in \mathbb{R}^+$, Lagrange multiplier update rate $\mu \in \mathbb{R}^+$, starting step $s$ for CBR.
**Require:** Optimizer $\mathrm{Opt}(\cdot)$ for minimization, Regularization function $R(\boldsymbol{\theta})$ (e.g. L2-norm)

1: # Initialization
2: $t \leftarrow 0$
3: $\boldsymbol{\theta}_t \leftarrow \mathrm{Initialize}(L(\cdot))$
4: $\lambda_t^j \leftarrow 0$ for $j = 1, \cdots, J$
5: $\kappa^j \leftarrow \infty \; j = 1, \cdots, J$
6: # Training
7: **for** $\boldsymbol{X}_t, \boldsymbol{y}_t \sim \mathcal{D}$ **do**
8:      $\boldsymbol{\theta}_{t+1} \leftarrow \boldsymbol{\theta}_t + \mathrm{Opt}(L(\boldsymbol{\theta}_t, \boldsymbol{X}_t, \boldsymbol{y}_t), \eta)$          $\triangleright$ Classic parameter update using, e.g., Adam.
9:      **for** each regularized parameter group $\boldsymbol{\theta}_t^j$ in $\boldsymbol{\theta}_t$ **do**
10:          $\lambda_{t+1}^j \leftarrow \left( \lambda_t^j + \mu \cdot (R(\boldsymbol{\theta}_t^j) - \kappa^j) \right)^+$
11:          $\boldsymbol{\theta}_{t+1}^j \leftarrow \boldsymbol{\theta}_{t+1}^j - \nabla_{\boldsymbol{\theta}^j} R(\boldsymbol{\theta}_t^j) \cdot \lambda_{t+1}^j$
12:          **if** $t = s$ **then**          $\triangleright$ Kappa-kI$_s$ initialization, see Section 4.2.
13:             $\kappa^j \leftarrow R(\boldsymbol{\theta}_t^j)$
14:          **end if**
15:      **end for**
16:      $t \leftarrow t + 1$
17: **end for**

---

## C  THE ADACPR ALGORITHM

---

**Algorithm 3** Optimization with adaptive bound constrained parameter regularization ( Ada CPR ).

---

**Require:** Loss Function $L(\boldsymbol{\theta}, \boldsymbol{X}, \boldsymbol{y})$ with parameters $\boldsymbol{\theta}$, and data $\mathcal{D} = \{(\boldsymbol{X}_n, \boldsymbol{y}_n)\}_{n=0}^N$
**Require:** Hyperparameters: Learning rate $\eta \in \mathbb{R}^+$, Lagrange multiplier update rate $\mu \in \mathbb{R}^+$
**Require:** Optimizer $\mathrm{Opt}(\cdot)$ for minimization, Regularization function $R(\boldsymbol{\theta})$ (e.g. L2-norm)
 1: # Initialization
 2: $t \leftarrow 0$
 3: $\boldsymbol{\theta}_t \leftarrow \mathrm{Initialize}(L(\cdot))$
 4: $\lambda_t^j \leftarrow 0$ for $j = 1, \cdots, J$
 5: $\kappa^j \leftarrow \boldsymbol{\theta}_t^j - \mathrm{Initialize}(\boldsymbol{\theta}_0^j)$ for $j = 1, \cdots, J$
 6: # Training
 7: **for** $\boldsymbol{X}_t, \boldsymbol{y}_t \sim \mathcal{D}$ **do**
 8:     $\boldsymbol{\theta}_{t+1} \leftarrow \boldsymbol{\theta}_t + \mathrm{Opt}(L(\boldsymbol{\theta}_t, \boldsymbol{X}_t, \boldsymbol{y}_t), \eta)$     ▷ Classic parameter update using, e.g., Adam.
 9:     **for** each regularized parameter group $\boldsymbol{\theta}_t^j$ in $\boldsymbol{\theta}_t$ **do**
10:         $\lambda_{t+1}^j \leftarrow \left(\lambda_t^j + \mu \cdot (R(\boldsymbol{\theta}_t^j) - \kappa^j)\right)^+$
11:         $\boldsymbol{\theta}_{t+1}^j \leftarrow \boldsymbol{\theta}_{t+1}^j - \nabla_{\boldsymbol{\theta}^j} R(\boldsymbol{\theta}_t^j) \cdot \lambda_{t+1}^j$
12:         **if** $\lambda_t^j = 0$ and $\lambda_{t-1}^j > 0$ **then**     ▷ Update $\kappa^j$ if the constraints are not active.
13:             $\kappa^j \leftarrow R(\boldsymbol{\theta}_t^j)$
14:         **end if**
15:     **end for**
16:     $t \leftarrow t + 1$
17: **end for**

---

# D EXPERIMENTS ON UPDATE RATE $\mu$

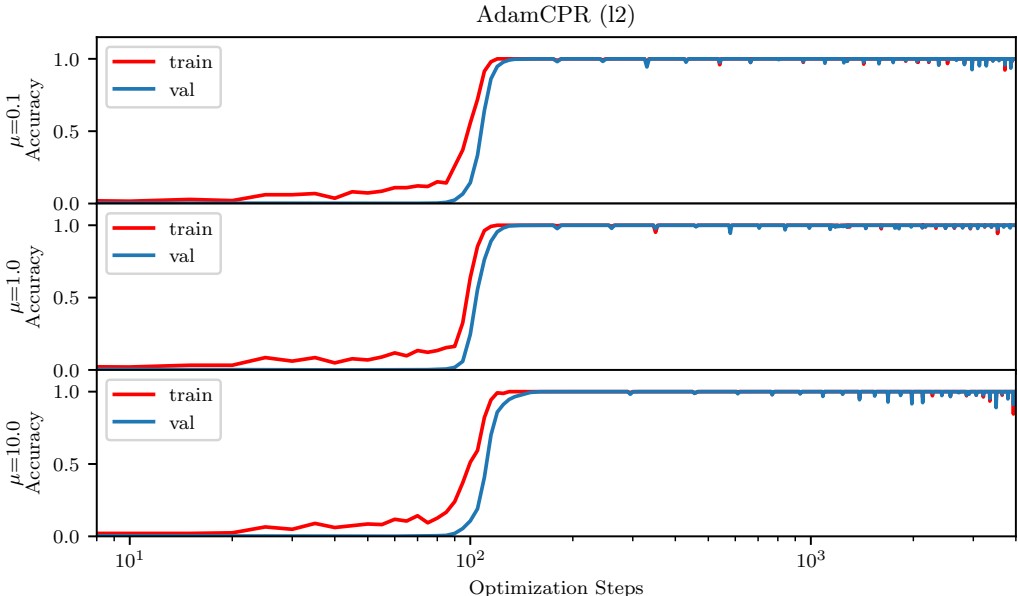

Figure D.1: Experiments on the modular addition task to illustrate the impact of different update rates $\mu$ on CPR. On the x-axis are the training steps in log scale, the red line represents the training, and the blue line the validation accuracy. We train AdamCPR with `Kappa-kI`$_0$ = 0.8 and on an $L_2$ norm with different values of the $\mu$ parameter. The experiments show the little impact of the $\mu$ parameter.

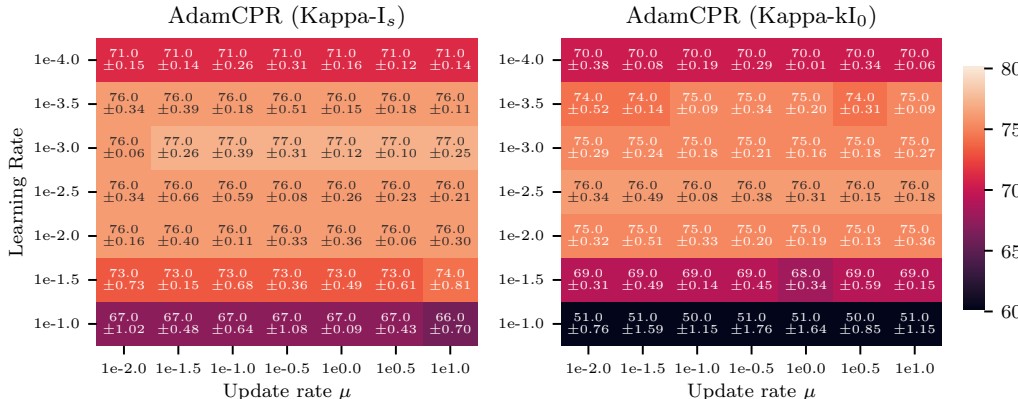

Figure D.2: The Figure shows the percentage of correct labels of the ResNet18 trained on the CIFAR100 with the use of `Kappa-kI`$_0$ (left), AdamCPR (`Kappa-I`$_s$) (right) with different update rates $\mu$. The elements in the heat map are experiments with different learning rates and each element is colored according to the mean accuracy of three random seeds and the numbers are the mean accuracy and standard deviation of the experiments. The experiment shows that the AdamCPR regularization is not sensitive to the choice of the $\mu$ parameter.

# E EXPERIMENTS ON MODULAR ADDITION TASK

Table E.1: Hyperparameters in the modular addition task.

| Parameter | Value |
| --- | --- |
| Modular addition p-value | 113 |
| Train fraction | 0.3 |
| Batch size | 512 |
| Model dim | 128 |
| Number of layers | 1 |
| Number of heads | 4 |
| Activation | ReLU |
| Initialization type | sqrt_dim |
| Learning rate | 0.001 |
| Adam $\beta_1$ | 0.9 |
| Adam $\beta_2$ | 0.98 |
| Exclude from regularization | bias, norm |

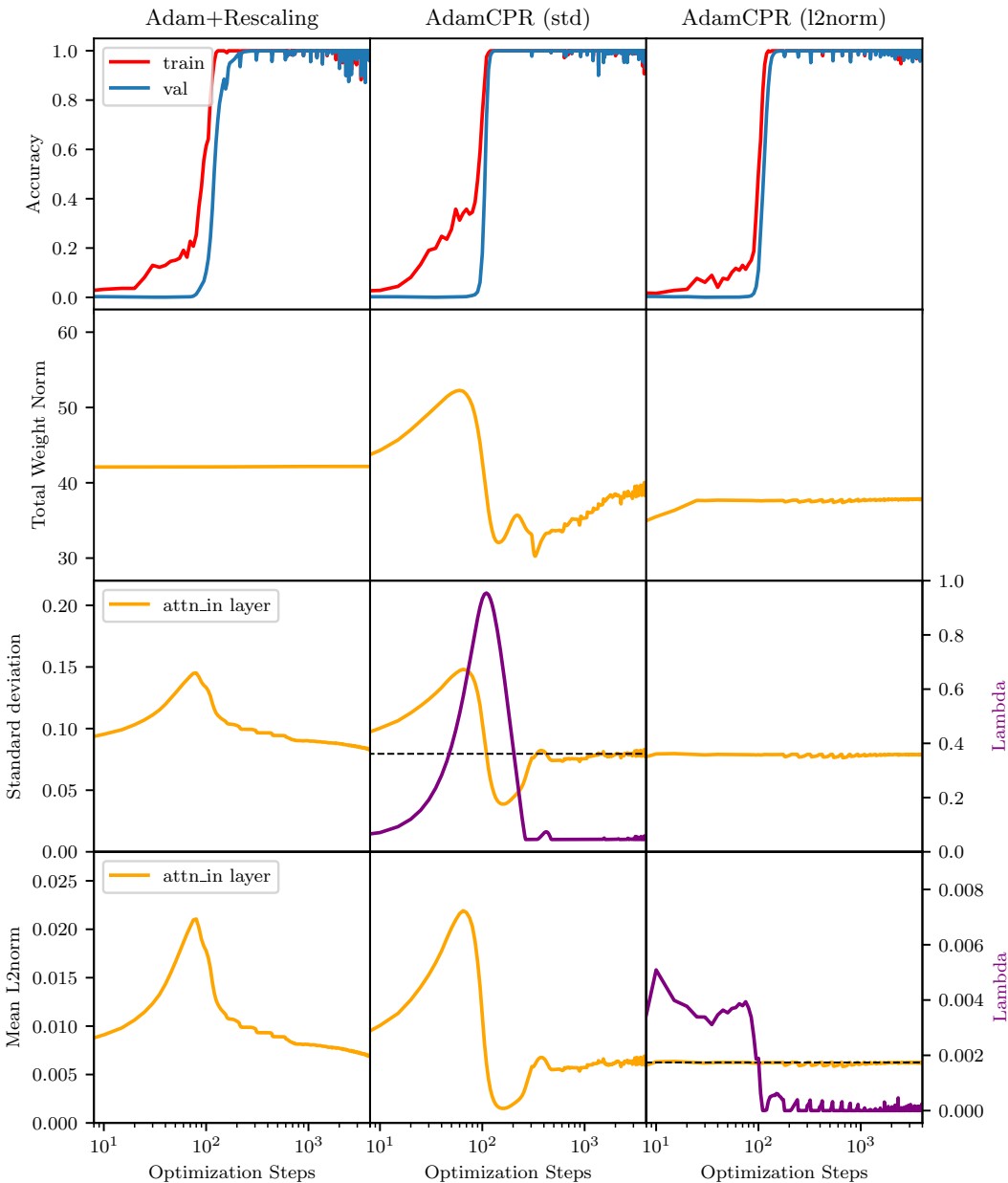

Figure E.1: Experiments on the modular addition task to illustrate the effect of rescaling and CPR, on the x-axis are the training steps in log scale, in the top row is red the training and blue the validation accuracy. In the second row, we see the total weight norm, below, we see the standard deviation of the *attention-in* weight parameter during the training progress and at the bottom the mean $L_2$ norm respectively. In the left column, we use Adam plus rescaling to a total weight norm of $0.8$ for optimization with a weight decay of $1.0$, and in the middle and right columns Adam with CPR. We optimize in the middle to a standard deviation of $0.9$ times the initial value and on the right to an $L_2$ norm of $0.8$ times the initial value.

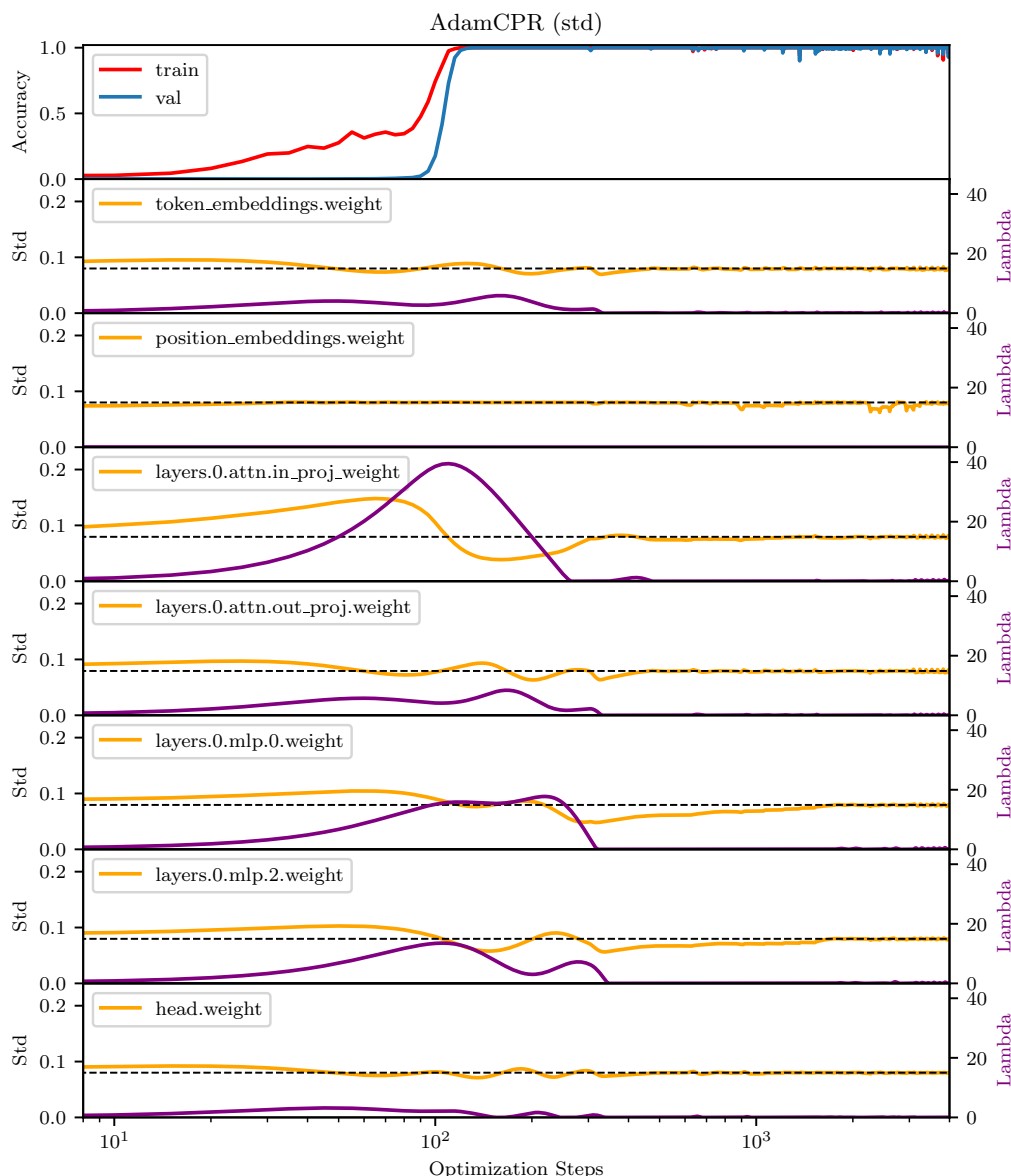

Figure E.2: Experiments on the modular addition task to illustrate the effect of CPR with regularization on the standard deviation on the different layers in the neural network. The x-axis displays the training steps in log scale. The top row shows the training and the validation accuracy in red and blue respectively. In the rows below, we see the standard deviation of the different layers during the training progress.

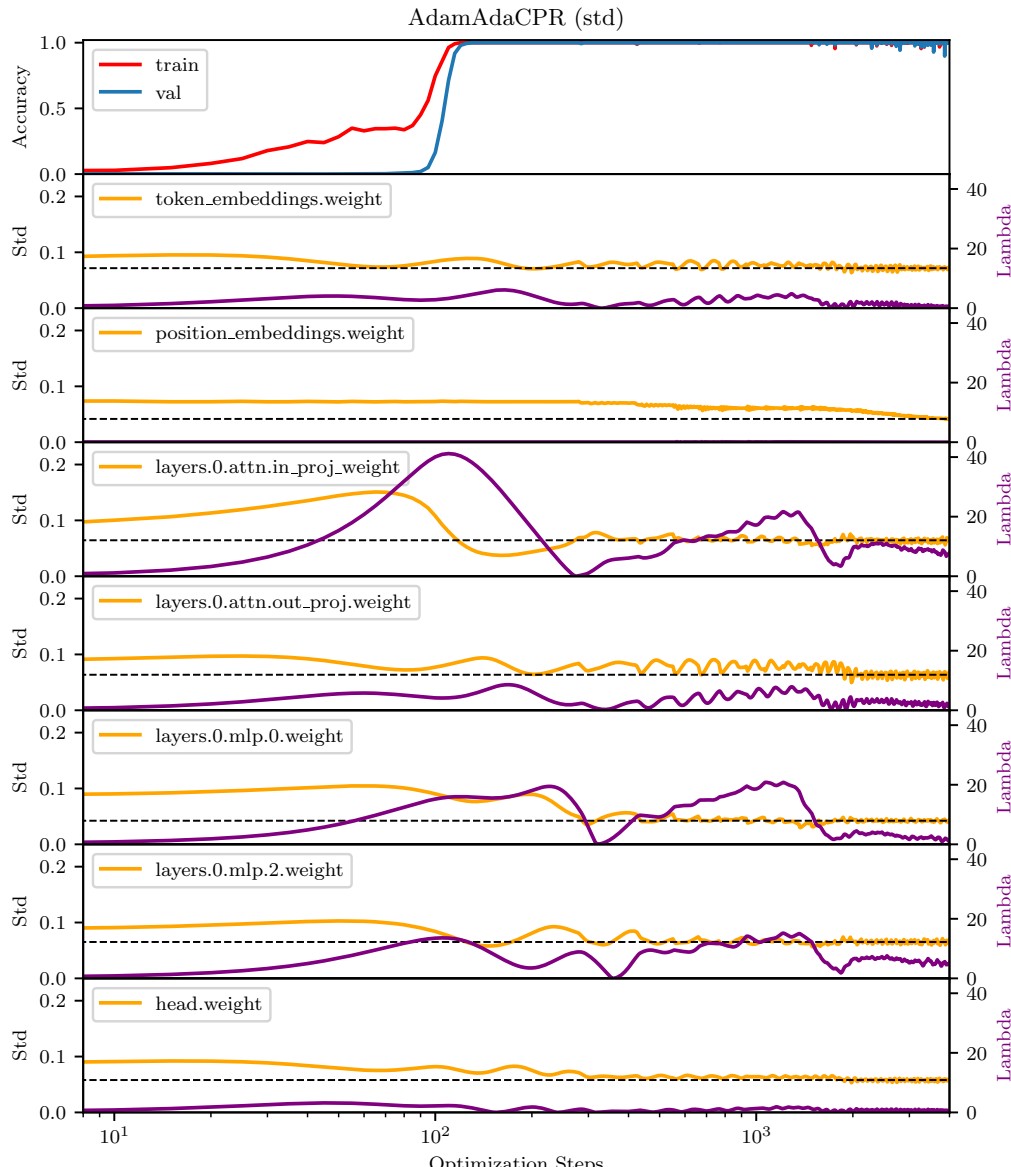

Figure E.3: Experiments on the modular addition task to illustrate the effect of AdaCPR with regularization on the standard deviation on the different layers in the neural network. The x-axis displays the training steps in log scale. The top row shows the training and the validation accuracy in red and blue respectively. In the rows below, we see the standard deviation of the different layers during the training progress.

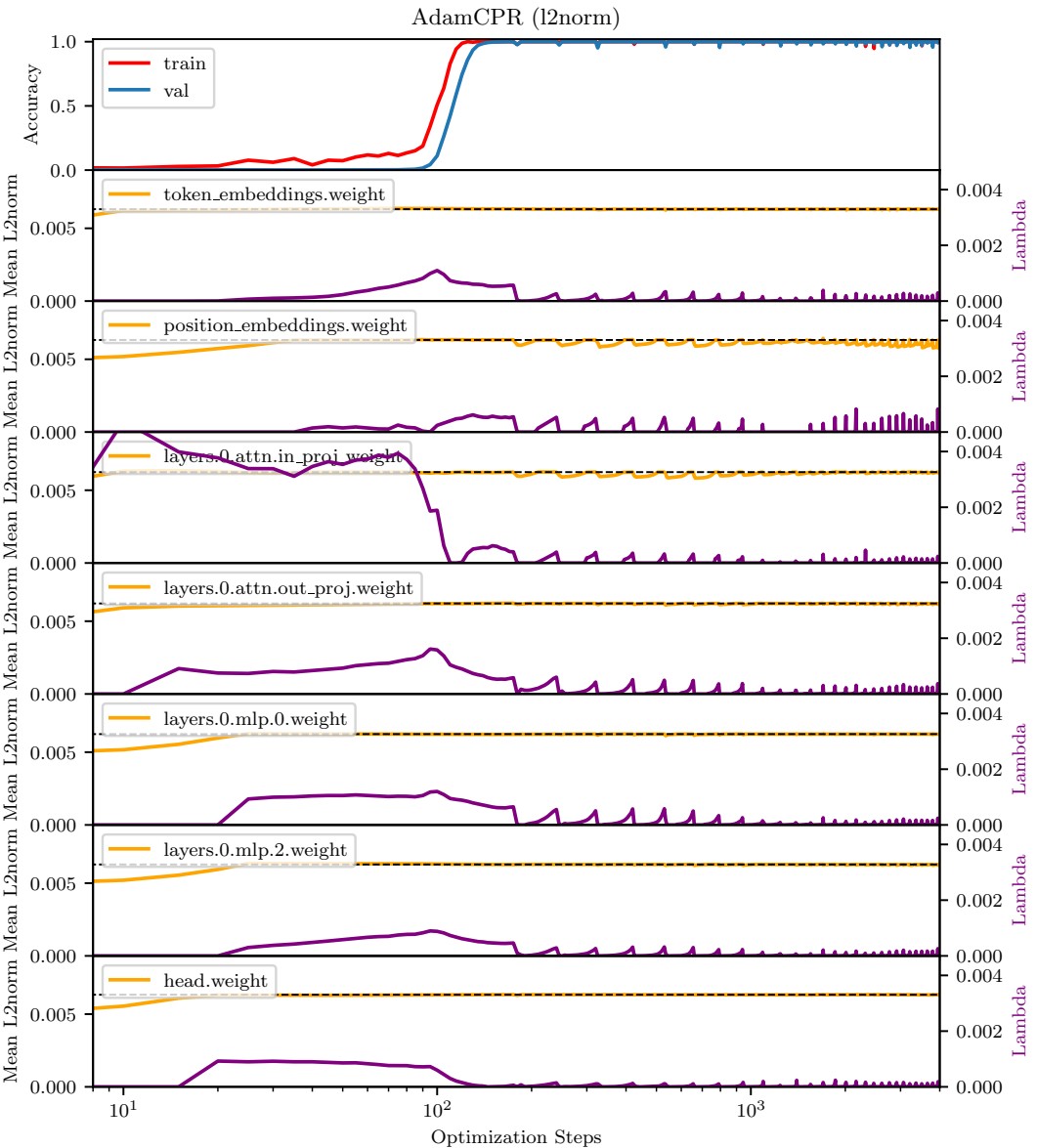

Figure E.4: Experiments on the modular addition task to illustrate the effect of CPR with regularization on the standard deviation on the different layers in the neural network. The x-axis displays the training steps in log scale. The top row shows the training and the validation accuracy in red and blue respectively. In the layer below, we see the mean $L_2$ norm of the different layers during the training process.

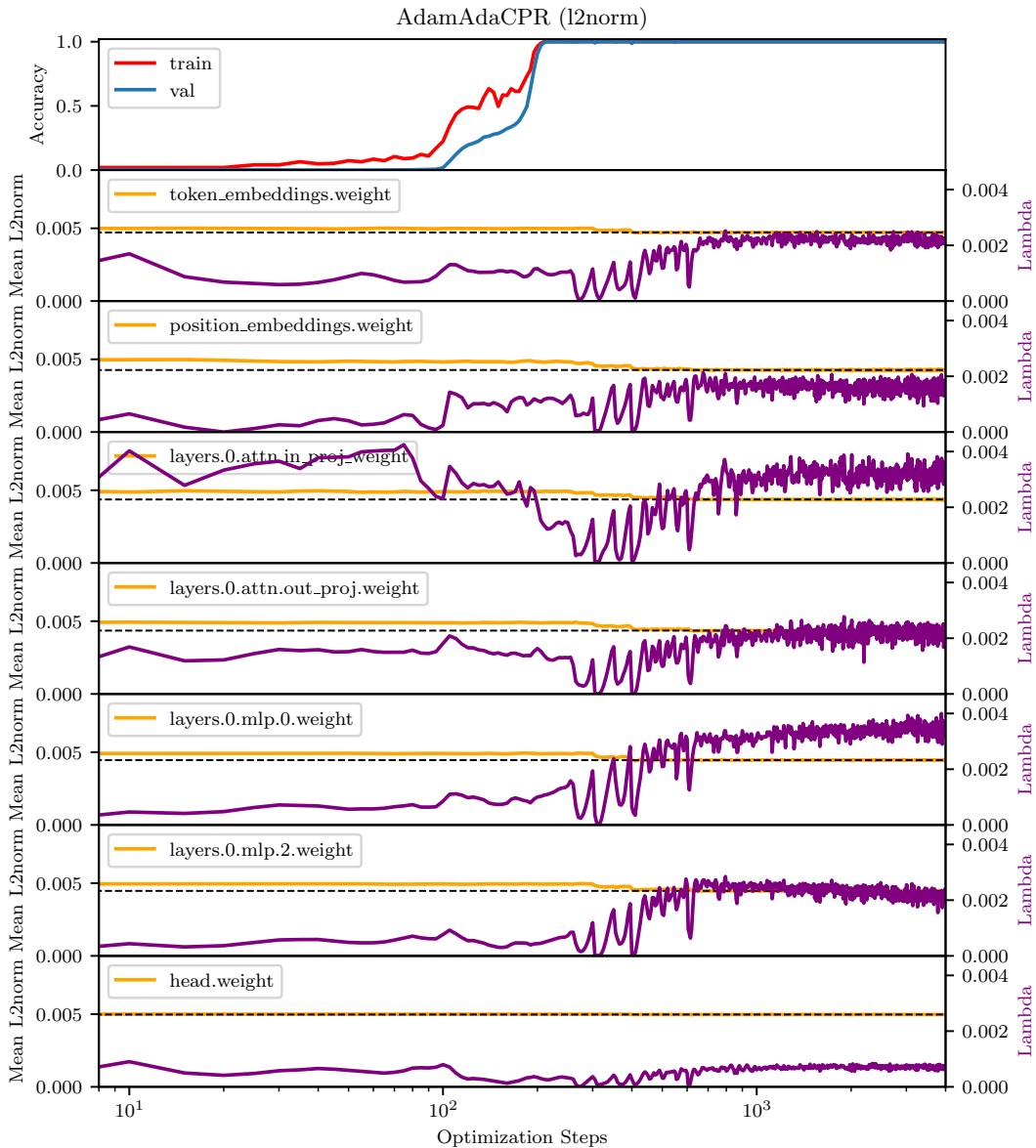

Figure E.5: Experiments on the modular addition task to illustrate the effect of AdaCPR with regularization on the standard deviation on the different layers in the neural network. The x-axis displays the training steps in log scale. The top row shows the training and the validation accuracy in red and blue respectively. In the rows below, we see the mean $L_2$ norm of the different layers during the training process.

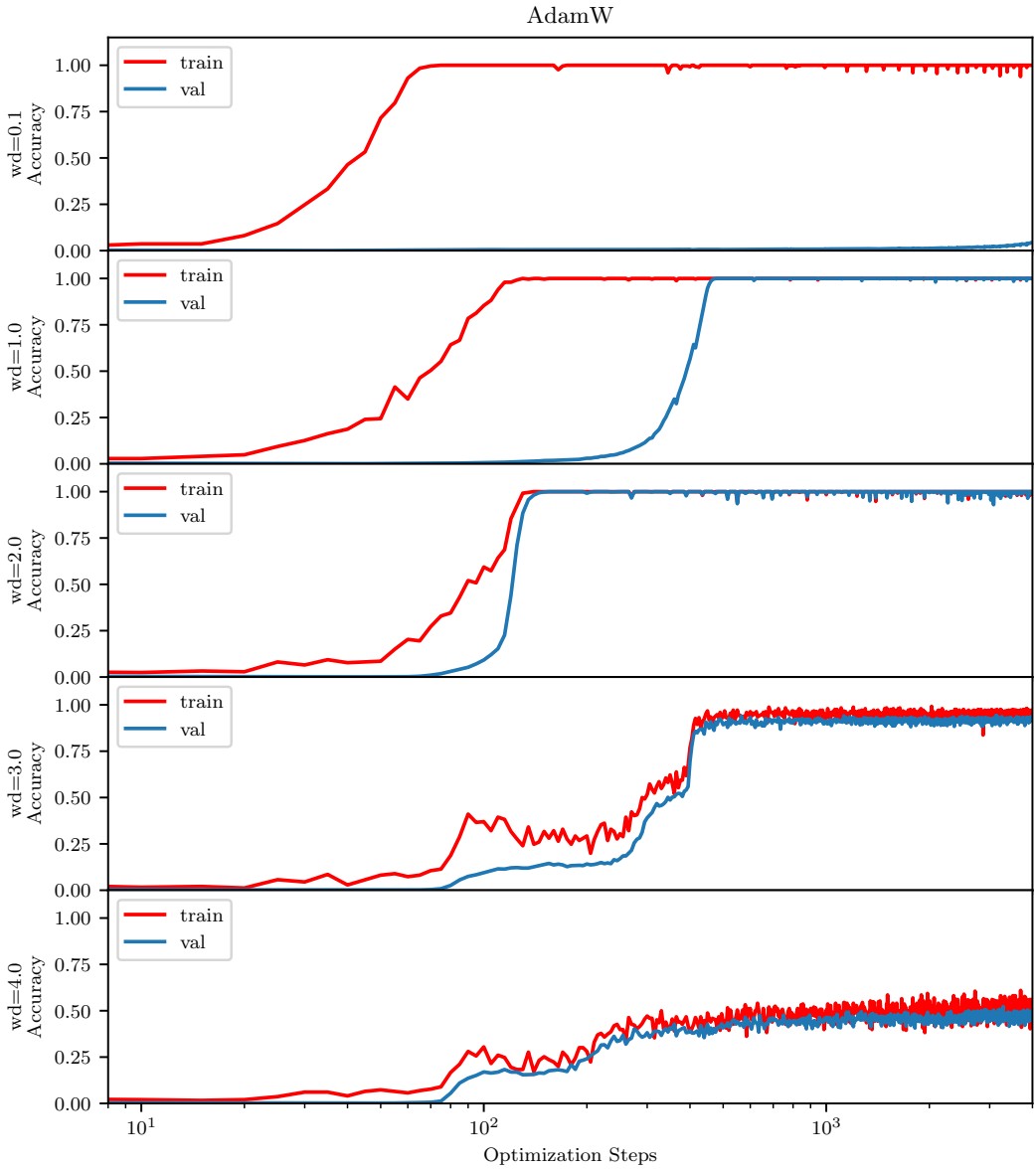

Figure E.6: Experiments on the modular addition task to illustrate the effect of weight decay in AdamW. We need a relatively high weight decay parameter to see the grokking phenomena but when we keep increasing the weight decay parameter, we can close the gap even more but it starts to get unstable after a value of 2.

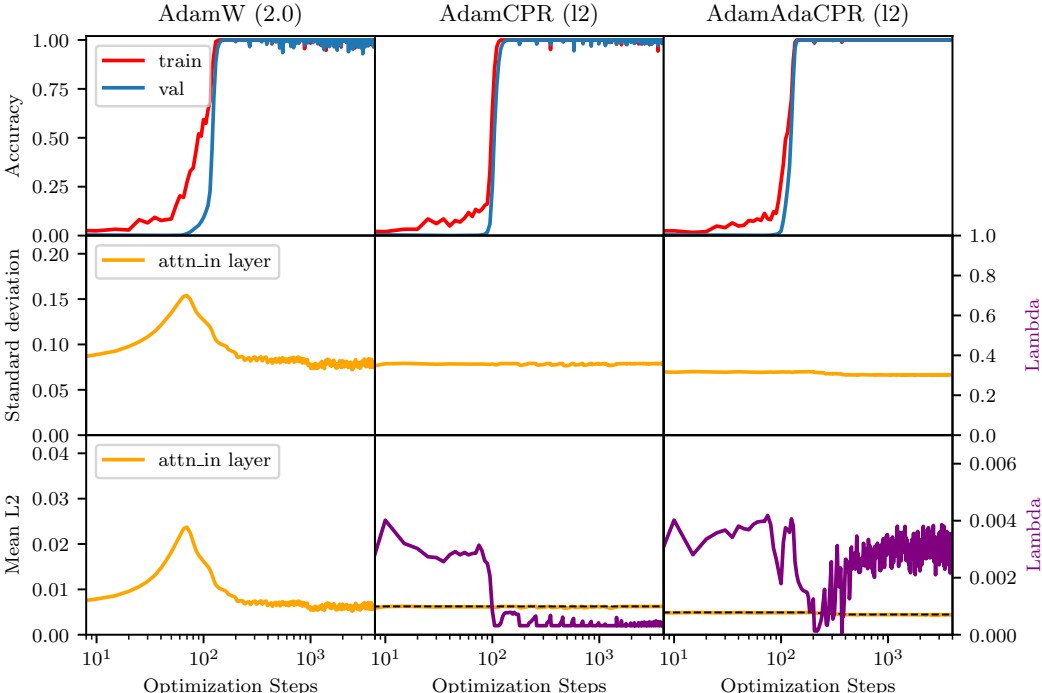

Figure E.7: Experiments on the modular addition task to illustrate the effect of weight decay in AdamW in comparison to AdamCPR and AdamAdaCPR (both Kappa-I$_s$, l2). We see that a high weight decay value reduces the gap between training and validation performance but comes with a higher instability in comparison to AdamCPR and AdamAdaCPR.

# F EXPERIMENTS ON IMAGE CLASSIFICATION

Table F.1: Hyperparameters of the ResNet18 on CIFAR100 experiment.

| Parameter | Value |
|---|---|
| Seed | 1,2,3 |
| Dataset | CIFAR100 |
| Batch size | 128 |
| Training Steps | 20000 |
| Model | ResNet18 |
| Optimizer | AdamW / Adam+Rescaling / AdamCPR |
| Learning Rate | 0.001 |
| Beta1 | 0.9 |
| Beta2 | 0.98 |
| Weight Decay | 0.1 |
| Lr Schedule | Cosine with warmup |
| Lr Warmup Steps | 500 |
| Lr Decay Factor | 0.1 |
| Rescale Alpha | $0, 0.8 \ldots 16$ |
| CPR$-\mu$ | 1.0 |
| CPR-$\kappa$ | $0.8 \ldots 16$ |
| CPR-$k$ | $4 \ldots 256$ |
| CPR-$\kappa$ warm-start steps | $250 \ldots 16000$ |
| Adaptive Bounds | False / True |

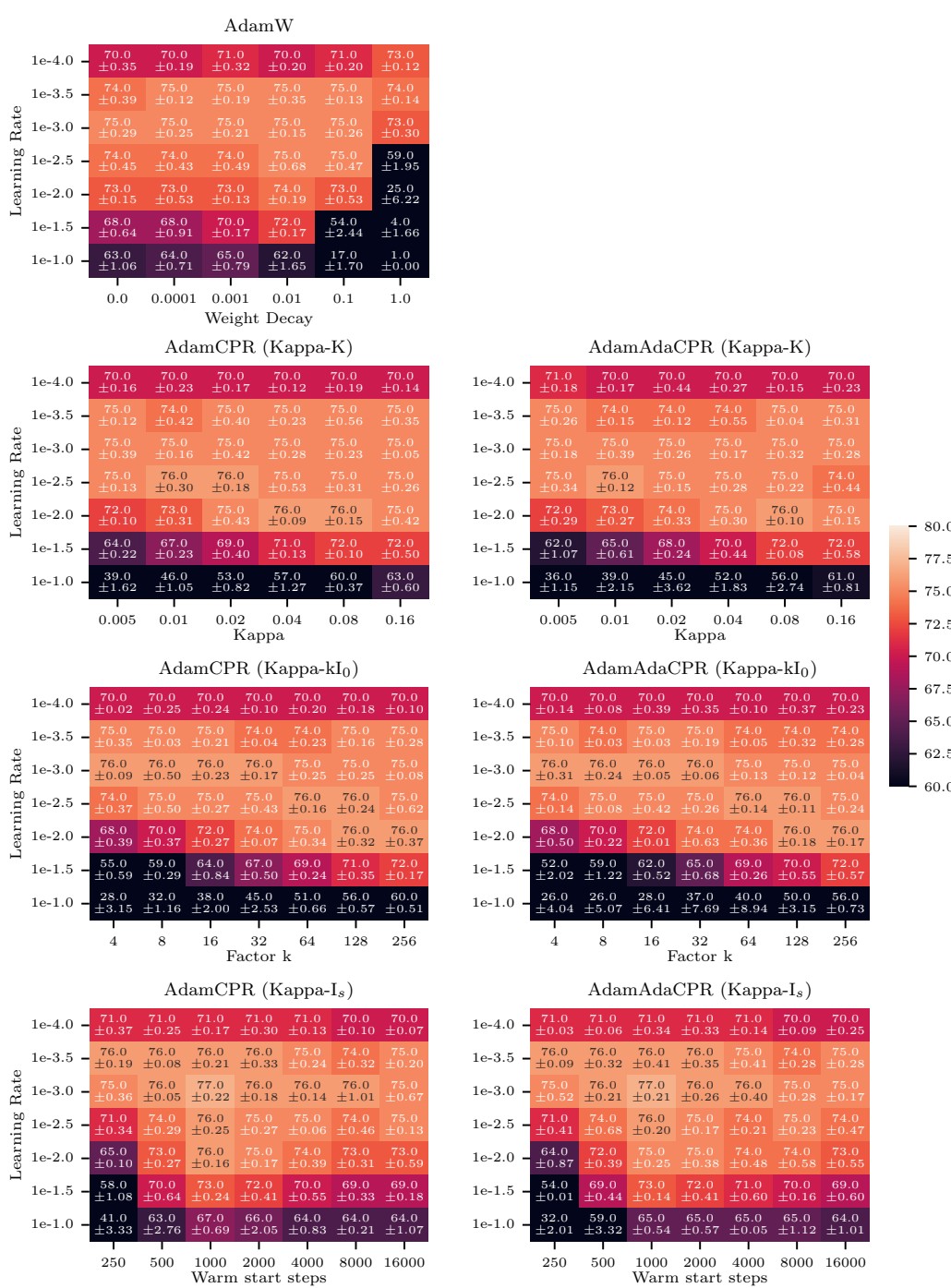

Figure F.1: Percentage of correct labels of the ResNet18 trained on the CIFAR100 with use of AdamW (top left), and below Adam with CPR (left) and AdaCPR (right) with use of the three different initialization techniques from Section 4.2, from top to bottom: Kappa-K, Kappa-kI$_0$, and Kappa-I$_s$. The elements in the heat map are experiments with different learning rates and regularization hyperparameters. Each element is colored according to the mean accuracy of three random seeds and the numbers are the mean accuracy and standard deviation of the experiments.

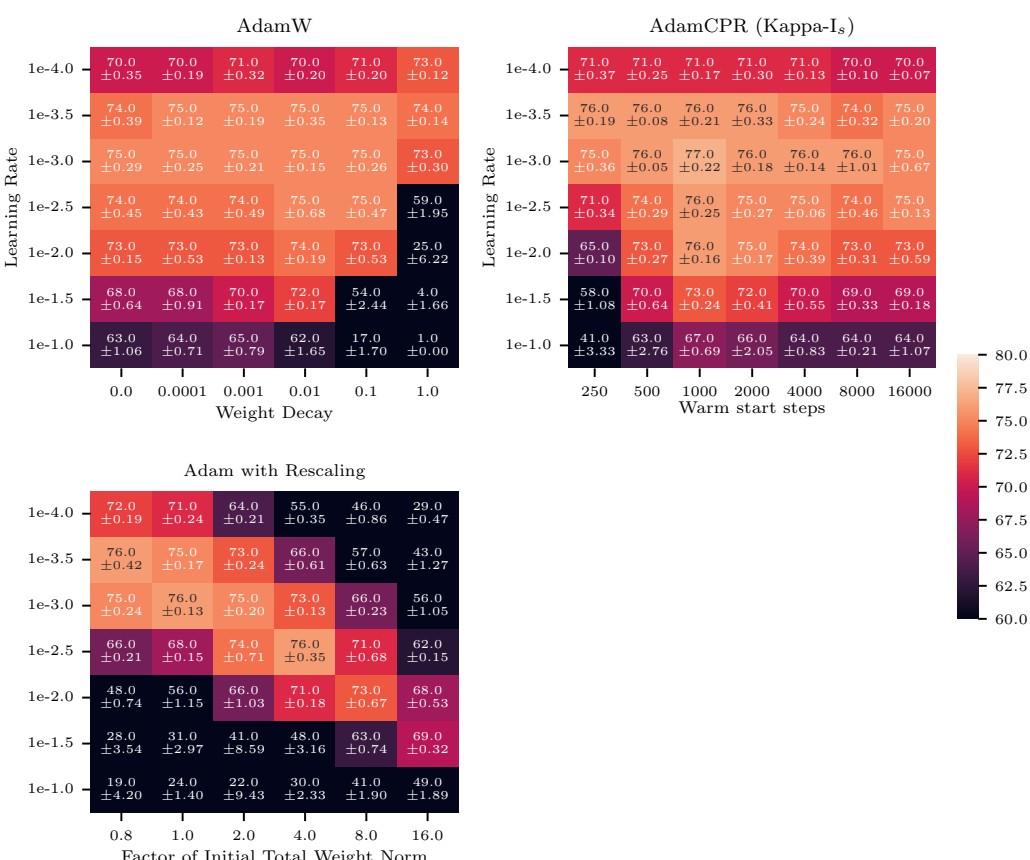

Figure F.2: Comparison of AdamW, AdamCPR, and Rescaling. The Figure shows the percentage of correct labels of the ResNet18 trained on the CIFAR100 with the use of AdamW (top left), AdamCPR (Kappa-$I_s$) (top right), and Adam with Rescaling with different factors of the initial total weight norm (bottom left). The elements in the heat map are experiments with different learning rates and regularization hyperparameters. Each element is colored according to the mean accuracy of three random seeds and the numbers are the mean accuracy and standard deviation of the experiments.

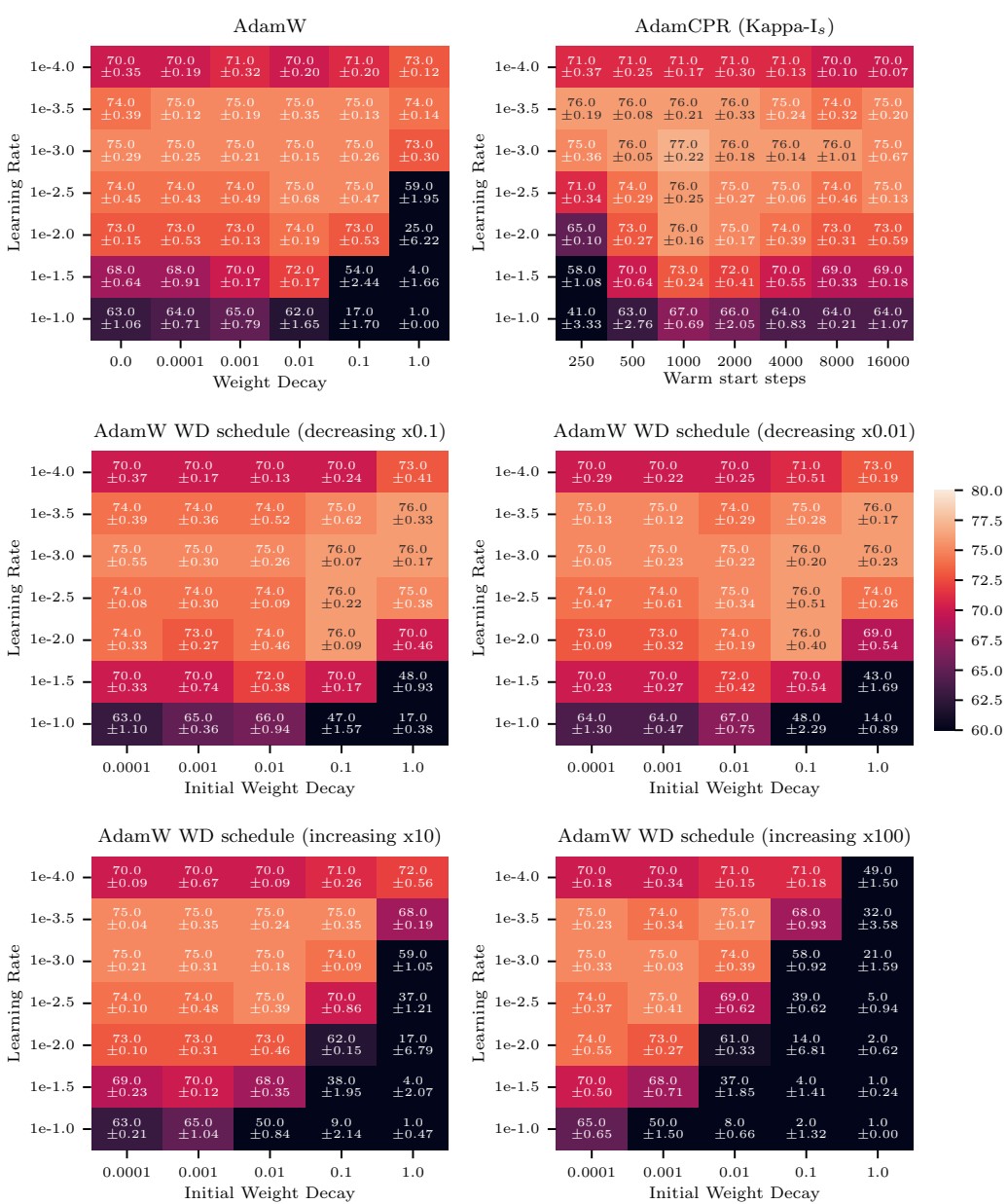

Figure F.3: Comparison of AdamW, AdamCPR, and weight decay scheduling similar to Caron et al. (2021); Oquab et al. (2023). The Figure shows the percentage of correct labels of the ResNet18 trained on the CIFAR100 with the use of AdamW (top left), AdamCPR (Kappa-I$_s$) (top right) and Adam with weight decay scheduling. We evaluated the task with cosine decreasing weight decay to 0.1 and 0.01 times of the initial weight decay value and with cosine increasing weight decay to 10 and 100 times of the initial weight decay value. The elements in the heat map are experiments with different learning rates and regularization hyperparameters. Each element is colored according to the mean accuracy of three random seeds and the numbers are the mean accuracy and standard deviation of the experiments.

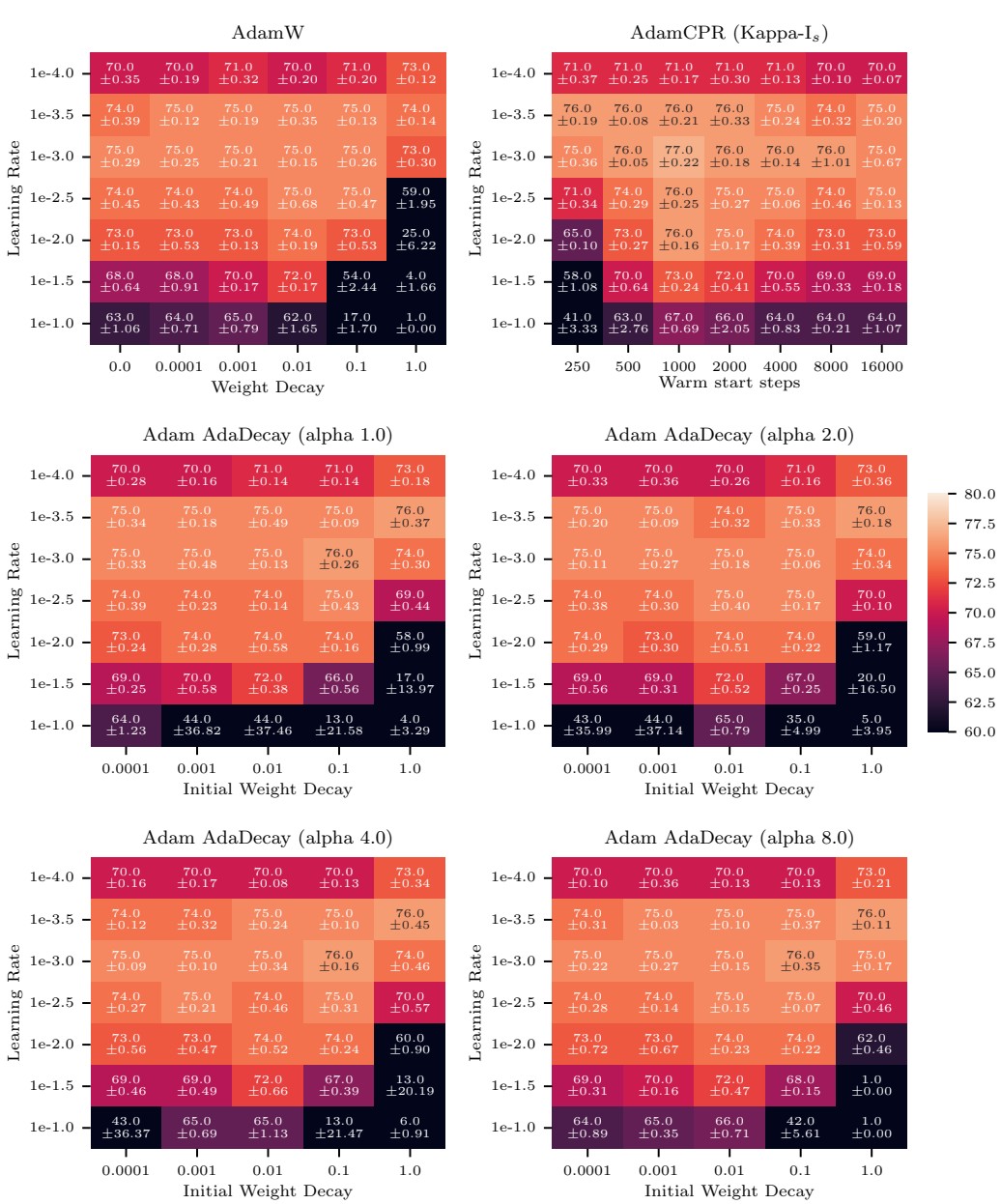

Figure F.4: Comparison of AdamW, AdamCPR, and Adam with AdaDecay Nakamura & Hong (2019). The Figure shows the percentage of correct labels of the ResNet18 trained on the CIFAR100 with the use of AdamW (top left), AdamCPR (Kappa-I$_s$) (top right), and Adam with AdaDecay with different (1.0, 2.0, 4.0, 8.0) values for the alpha hyperparameter in AdaDecay. The elements in the heat map are experiments with different learning rates and regularization hyperparameters. Each element is colored according to the mean accuracy of three random seeds and the numbers are the mean accuracy and standard deviation of the experiments.

# G    EXPERIMENTS ON LANGUAGE MODELLING

Table G.1: The mean results of three random seeds and the corresponding standard deviation of the GPT2s training over 200k steps. The rows denote the weight decay factor $\gamma$ for AdamW, and for CPR and AdaCPR, they indicate the number of warm-start steps $s$ of the initialization Kappa-I$_s$. The $L_2$ norm is used as a regularization function.

| Method | | Accuracy ↑ | PPL ↓ |
|---|---|---|---|
| AdamW | 1e-3 | $0.445 \pm 0.0004$ | $17.98 \pm 0.04$ |
| **AdamW** | **1e-2** | $0.446 \pm 0.0003$ | $17.84 \pm 0.05$ |
| AdamW | 1e-1 | $0.441 \pm 0.0004$ | $18.58 \pm 0.04$ |
| CPR | 5k | $0.445 \pm 0.0005$ | $17.96 \pm 0.07$ |
| **CPR** | **10k** | $\mathbf{0.447 \pm 0.0001}$ | $\mathbf{17.68 \pm 0.02}$ |
| CPR | 20k | $0.446 \pm 0.0003$ | $17.80 \pm 0.03$ |
| AdaCPR | 5k | $0.445 \pm 0.0006$ | $17.95 \pm 0.02$ |
| AdaCPR | 10k | $0.447 \pm 0.0005$ | $17.69 \pm 0.04$ |
| AdaCPR | 20k | $0.446 \pm 0.0003$ | $17.79 \pm 0.02$ |

Table G.2: We evaluated the best performing AdamW weight decay value against the best performing AdamCPR configuration on a long run GPT2s training (400k steps) and on the larger model GPT2m. We provide the mean accuracy and PPL as well as the standard deviation across three seeds.

| GPT2s 400k | AdamW 1e-2 | AdamCPR 10k |
|---|---|---|
| Accuracy ↑ | $0.449 \pm 0.0003$ | $\mathbf{0.450 \pm 0.0001}$ |
| PPL ↓ | $17.43 \pm 0.04$ | $\mathbf{17.32 \pm 0.02}$ |

| GPT2m 200k | AdamW 1e-2 | AdamCPR 10k |
|---|---|---|
| Accuracy ↑ | $0.472 \pm 0.0002$ | $\mathbf{0.474 \pm 0.0002}$ |
| PPL ↓ | $14.23 \pm 0.02$ | $\mathbf{14.03 \pm 0.02}$ |

Table G.3: Hyperparameters of the language modeling task (GPT2 and Openwebtext).

| Parameter | GPT2s 200k | GPT2s 400k | GPT2m |
|---|---|---|---|
| GPUs | | 8 | |
| Gradient Clip Val | | 1.0 | |
| Max Steps | 200k | 400k | 200k |
| Precision | | bf16-mixed | |
| Seed | | 1234 | |
| Optimizer | | AdamW or AdamCPR | |
| Learning Rate | | 0.002 | |
| Weight Decay | | 0.1 | |
| Beta1 | | 0.9 | |
| Beta2 | | 0.99 | |
| Eps | | $1.0 \times 10^{-9}$ | |
| Stat Measure | | $L_2$ norm | |
| Kappa | | None, $0.005 \cdots 0.16$ | |
| Kappa Factor | | False, $4 \cdots 256$ | |
| Lagmul Rate | | 1.0 | |
| Kappa Adapt | | True / False | |
| Kappa Init After Steps | | False, $250 \cdots 16k$ | |
| Bias Weight Decay | | False | |
| Normalization Weight Decay | | False | |
| Lr Num Warmup Steps | | 4000 | |
| Lr Decay Factor | | 0.1 | |
| Lr Schedule | | Cosine | |
| Deepspeed Stage | | 2 | |
| Model Dimension | 768 | 768 | 1024 |
| Number of Layers | 12 | 768 | 24 |
| Number of Heads | 12 | 12 | 16 |
| Fed Forward Dim | 3072 | 3072 | 4048 |
| Attn Dropout | | 0.1 | |
| Resi Dropout | | 0.1 | |
| Embed Dropout | | 0.1 | |
| Rotary Pos Embed | | True | |
| Rotary Emb Fraction | | 0.5 | |
| Softmax Scale | | True | |
| Key Dim Scaler | | True | |
| Gating | | False | |
| Use Glu | | False | |
| Use Bias | | True | |
| Flash Attn | | True | |
| Initializer | | Xavier Uniform | |
| Dataset Name | | Openwebtext | |
| Max Sample Len | | 1024 | |
| Batch Size | 32 | 32 | 24 |
| Val Ratio | | 0.0005 | |

# H  EXPERIMENTS ON MEDICAL IMAGE SEGMENTATION

Table H.1: Hyperparameters of the medical image segmentation experiments.

| Parameter | Value |
| --- | --- |
| Fold | 0,1,2,3,4 |
| Dataset | BTCV, Heart, BraTS |
| Preprocessing | Default nnU-Net preprocessing (Isensee et al., 2021) |
| Batch size | 2 (following (Isensee et al., 2021) |
| Patch size | (48x192x192) BTCV, (80x192x160) Heart, (128x128x128) BraTS |
| Training Steps | 125k (BTCV), 25k (Heart &BraTS) |
| Model | 3d fullres U-Net (following (Isensee et al., 2021)) |
| Optimizer | AdamW / AdamCPR |
| Learning Rate | 0.01 |
| Beta1 | 0.9 |
| Beta2 | 0.99 |
| Weight Decay | $1e-5\dots1e-1$ (AdamW) |
| Lr Schedule | Polynomial decay with warmup |
| Lr Warmup Steps | 2000 |
| Lr Polynomial exponent | 0.9 |
| CPR-$\mu$ | 1.0 |
| CPR-$\kappa$ | 1.0 |
| CPR-$k$ | False |
| CPR-$\kappa$ warm-start steps | $1000\dots4000$ |
| Adaptive Bounds | False |

