# OpenReview forum: "Constrained Parameter Regularization"
_ICLR.cc/2024/Conference — Submitted to ICLR 2024_

### Official Review · Reviewer_EQkA · 2023-10-29

**Soundness:** 3 good
**Presentation:** 2 fair
**Contribution:** 2 fair
**Rating:** 5
**Confidence:** 2

**Summary:**

The paper proposes a constrained parameter regularization (CPR) technique that dynamically adjusts its regularization strength on parameters based on the violation of the constraints. The authors address the optimization of CPR by an adaptation of the augmented Lagrangian method and provide a simple mechanism to adapt the upper bounds during the optimization. The experimental results speak in favor of the proposed method.

**Strengths:**

- The paper studies a very important field as the performance of modern NNs are often bottlenecked by the underlying optimization and many tricks boil down to easing the optimization difficulty. I found the topic significant and interesting.
- The design of adjusting the regularization strength dynamically is very interesting and novel.
- The performance of the proposed method looks good in terms of both the average and the best performance. The method is also robust against different design choices of $\kappa$.

**Weaknesses:**

- Although it is a very interesting paper, I feel like the analysis is not good enough. There are many interesting aspects that could be further explored IMHO. For example:
1. If $\lambda$ is used to accumulate constraint violations, shouldn't it be $\lambda_{t+1}=\lambda_t + \mu(R(\theta_t) - \kappa)^+$?
2. Have the authors considered adjusting $\lambda$ in a non-linear way? For example $\lambda_{t+1}=\lambda_t + \mu f((R(\theta_t) - \kappa))^+$, where $f$ is a non-linear function. Maybe it leads to faster convergence if tuned properly?
3. Should $\lambda$ have a self decay factor? Similar to a diverging behavior with large learning rates, when a feasible point $\theta_{t}$ is found at a very large $\lambda_t$. In the next iteration, $\theta_{t+1}$ may overshoot in the opposite direction and becomes infeasible again because $\lambda_t$ has not yet become small enough.

- The experimental setting is not very clear to me. What is the percentage of correct labels in the Object Detection experiment? Is the "Object Detection" actually classification on CIFAR100? As "Object Detection" is a well-established field itself, I found the terms very confusing.

- How significant is the performance difference between CPR and weight decay? As the absolute performance difference is not huge, it would be great if the standard deviation of different runs of the same setting can be provided.

- Maybe I missed it, but how about a baseline of weight decay factor dependent on the magnitude of the parameters, e.g. larger weights leads to a larger decay coefficient (linear or non-linear dependence)? I wonder how well this could achieve the goal of the proposed method.


Minor:
- Equation 6 should have $\kappa^j$ rather than $\kappa$
- The authors claim that the method is not sensitive to $\mu$, but it would still be great to see the analysis on it.
- Better to use consistent notations. E.g. in Figure E1, fix -> Kappa-K, dependent -> Kappa_kI0, warm started ->Kappa_Is
- Larger scale evaluation, e.g. on image-net, would be strongly advised.

**Questions:**

See weaknesses

---

> ### Author Response · Authors · 2023-11-14
> **Response to Reviewer EQkA 1/2**
>
> We thank the reviewer for the helpful feedback and hope our response and answers can convince the reviewer to increase their score. We address the concerns and questions as follows:
>
> **W1:** *If $lambda$  is used to accumulate constraint violations, shouldn't it be $\lambda_{t+1} = \lambda_t + \mu (R(\theta_t )-\kappa)^+ $?*
>
> Unfortunately, this wording was imprecise. $\lambda$ accumulates constraint function values (not constraint violations, otherwise $\lambda$ would never decrease). We adjusted the wording in the manuscript (Section 4.1).
>
>
> **W2:** *Have the authors considered adjusting $\lambda$  in a non-linear way? For example $\lambda_{t+1} = \lambda_t + \mu f ((R(\theta_t )-\kappa))^+ $, where $f$ is a non-linear function. Maybe it leads to faster convergence if tuned properly?*
>
> We have not tried a non-linear adjustment. The update rule is derived as the closed form solution for the maximization in $\hat{F}$, see Equation (2). Note that this can also be seen as a (projected) gradient ascent step on $f(x) + \lambda \cdot c(x)$ (see right side of Equation (1)) with respect to $\lambda$ and a learning rate $\mu$. Additionally, for a different choice of $\hat{F}$, a different update rule is obtained.
> Nevertheless, it should be noted that we use the method as a means to implement a computationally cheap adjustment procedure for the individual $\lambda_i$. It is not immediately clear if a faster convergence of the $\lambda_i$ would lead to more favorable training dynamics and thus an overall better training result.
>
>
> **W3:** *Should $\lambda$ have a self decay factor? Similar to a diverging behavior with large learning rates, when a feasible point $\theta_t$  is found at a very large $\lambda_t$. In the next iteration,  $\theta_{t+1}$  may overshoot in the opposite direction and becomes infeasible again because $\lambda_t$ has not yet become small enough.*
>
> If the constraint violations are growing moderately, and since we initialize $\lambda=0$, $\lambda$ should also not be growing too quickly. Additionally, the procedure seeks to find a stationary point where $\eta  \cdot \nabla L = - \lambda \cdot \nabla c$ (in case of gradient descent). Instabilities may arise, for example, if one of the balancing forces, in this case, the update direction for $L$ may drastically change. However, this should be unlikely when training with momentum on $L$.
> We thus assume that only a very unstable training progress could potentially cause such issues, and we never observed them in any of our experiments.
>
>
> **W4:** *The experimental setting is not very clear to me. What is the percentage of correct labels in the Object Detection experiment? Is the "Object Detection" actually classification on CIFAR100? As "Object Detection" is a well-established field itself, I found the terms very confusing.*
>
> We apologize for this misnaming. You are right, we mistakenly named the task “object detection” instead of “image classification”. We changed the wording in the paper accordingly to  “image classification”. The reported metric is simply accuracy.
>
> **W5:** *How significant is the performance difference between CPR and weight decay? As the absolute performance difference is not huge, it would be great if the standard deviation of different runs of the same setting can be provided.*
>
> We performed additional experiments for three random seeds in the language modeling experiment (similar to the image classification experiments) and added the standard deviation for experiments on image classification and language modeling. We added the standard deviation for the LLM experiments in the plot and in Tables 6 and 7 in Appendix G and for the image classification experiments in the heat map plots in Appendix F.
> To assess the significance of our results, we also performed a Welch’s t-test on the GPT2s experiments with the best AdamW and best AdamCPR configuration, showing a significant difference with a p-value of 0.018.
>
>
>
> **W6:** *Maybe I missed it, but how about a baseline of weight decay factor dependent on the magnitude of the parameters, e.g. larger weights leads to a larger decay coefficient (linear or non-linear dependence)? I wonder how well this could achieve the goal of the proposed method.*
>
> We are unsure if we understand this idea of a baseline correctly. Since the decoupled weight decay update already contains the parameter in the update step: $\theta_i = \theta_{i-1} - \eta \cdot \operatorname{Opt}(L)  - \eta \cdot \gamma \cdot \theta_{i-1}$. Should the weight decay value $\gamma$ additionally be a function of $\theta_0$ or  $\theta_{i-1}$?
>
> 1/2

---

> ### Author Response · Authors · 2023-11-14
> **Response to Reviewer EQkA 2/2**
>
> **WM1:** *Equation 6 should have $\kappa^j$  rather than $\kappa$*
>
> Thanks, yes indeed, we have fixed this missing index.
>
> **WM2:** *The authors claim that the method is not sensitive to $\mu$  but it would still be great to see the analysis on it.*
>
> We agree that the paper missed so far to show experiments on the update rate $\mu$. We added a new section in the Appendix (Appendix D) with two plots. One on the modular addition task and one on the image classification task which compares different $\mu$ values and demonstrates the low sensitivity of the update rate.
>
>
> **WM3:** *Better to use consistent notations. E.g. in Figure E1, fix -> Kappa-K, dependent -> Kappa_kI0, warm started ->Kappa_Is*
>
> Thanks for pointing us to this inconsistency, we corrected the figures accordingly.
>
> **WM4:** *Larger scale evaluation, e.g. on image-net, would be strongly advised.*
>
> We expected the GPT2m experiment to be a large-scale experiment since the GPT2m model has 354M parameters (in contrast to 124M in GPT2s). As far as we know, a ResNet152, which is often used for imagenet training, only has ~64M parameters.
>
> If we have addressed (some of) your concerns, we would be very thankful if you considered raising your score. We would be very happy to answer any follow-up questions and concerns.
>
> 2/2

---

> ### Author Response · Authors · 2023-11-19
>
> Dear Reviewer EQkA,
>
> We received a wide range of scores in our paper's reviews and have carefully addressed the feedback provided. We kindly request that you reconsider your evaluation in light of our responses. We have made specific revisions by correcting issues in our manuscript, adding a section about the update rule, and adding the standard deviation to provide significance. We believe these changes address your initial concerns.
>
> We welcome any further questions or the opportunity to discuss any remaining issues. Thank you for your time and consideration.

---

> ### Author Response · Authors · 2023-11-22
>
> Dear Reviewer EQkA,
>
> We submitted our rebuttal eight days ago and sent a reminder 3 days ago, and we would very much appreciate a reply during the reviewer-author discussion, which ends in 14 hours. We would gladly reply promptly if you still have any concerns or questions or replies. We‘re looking forward to your reply. Thank you for your service to ICLR!

---

### Official Review · Reviewer_HpN9 · 2023-11-01

**Soundness:** 2 fair
**Presentation:** 2 fair
**Contribution:** 1 poor
**Rating:** 3
**Confidence:** 5

**Summary:**

Summary： The paper proposes an alternative to traditional weight decay, which does not uniformly apply a constant penalty to all parameters. Instead, it enforces an upper bound on statistical measures for parameter groups and addresses this issue through adjustments to an enhanced Lagrangian method. It also allows for different regularization strengths for each parameter group, without the need to explicitly set penalty coefficients for regularization terms.

**Strengths:**

Strengths:
1） The authors introduce a constrained parameter regularization that does not uniformly
apply a constant penalty to all parameters.
2） The authors provide an open-source implementation of CPR, which can be easily adjusted
by replacing the optimizer class.

**Weaknesses:**

1） The authors should provide a clearer exposition of the problem statement and research
motivation in the abstract.
2） In the related work section, the authors should provide a more comprehensive review of
prior research.
3） The experimental results conducted by the authors did not show a significant
improvement in optimization performance, and they did not provide additional
experimental results or comparisons related to existing work.
4） While applying different regularization strengths to each parameter group may seem
intriguing, the authors have not discussed the necessity and potential advantages of this
approach compared to traditional methods.

**Questions:**

N/A

---

> ### Author Response · Authors · 2023-11-14
> **Response to Reviewer HpN9**
>
> We thank the reviewer for the helpful feedback and hope our response and answers can convince the reviewer to increase their score. We address the concerns and questions as follows:
>
> **W1:** *The authors should provide a clearer exposition of the problem statement and research motivation in the abstract.*
>
> We thank the reviewer for the constructive comments. We adjusted the abstract accordingly
> and tried to clearly emphasize the research question. We moved the motivation to the beginning of the abstract and reformulated it to have a clearer and more trenchant abstract.
>
> **W2:** *In the related work section, the authors should provide a more comprehensive review of prior research.*
>
> We added an additional related work on adaptive weight decay [Nakamura & Hong (2019)] and compared it experimentally. We also added baseline experiments on weight decay schedule and rescaling. Please find the experimental results in the appendix (Figures E2, E3, E4). To the best of our knowledge, we mentioned all relevant related work. Does the reviewer know any additional related work we missed so far? We would gladly add it.
>
>
> **W3:** *The experimental results conducted by the authors did not show a significant improvement in optimization performance, and they did not provide additional experimental results or comparisons related to existing work.*
>
> We performed additional experiments for three random seeds in the language modeling experiment and added the standard deviation for experiments on image classification and language modeling. We added the standard deviation for the LLM experiments in the plot and in Table G.6 and G.7 in Appendix G and for the image classification experiments in all heat map plots in Appendix F.
> To assess the significance of our results, we also performed a Welch’s t-test on the GPT2s experiments with the best AdamW and best AdamCPR configuration, showing a significant difference with a p-value of 0.018.
> Furthermore, we provide additional experiments for comparison to existing work (see comment on the previous point). Does this address the concerns of the reviewer regarding the significance and comparison to existing work?
>
>
> **W4:** *While applying different regularization strengths to each parameter group may seem intriguing, the authors have not discussed the necessity and potential advantages of this approach compared to traditional methods.*
>
> Choosing individual regularization strength is a more general case of choosing one strength for all parameter groups. Our hypothesis was that this additional flexibility leads to better performance. E.g., we do not regularize parameters that do not need regularization. We clarify this in the modified abstract too. In the introduction, we already write: "However, not all parameters in a neural network have the same role or importance and different weights could benefit from different regularizations. Similarly, it is unclear if a single weight decay value is optimal for the entire duration of optimization, especially for large-scale training.” Does this address your concern?
>
> If we have addressed (some of) your concerns, we would be very thankful if you considered raising your score. We would be very happy to answer any follow-up questions and concerns.

---

> ### Author Response · Authors · 2023-11-19
>
> Dear Reviewer HpN9,
>
> we received a wide range of scores in our paper's reviews and have carefully addressed the feedback provided. We kindly request that you consider our responses and would love to engage with you. We have made specific revisions by updating the abstract, adding the standard deviation to provide significance, and providing more related work and baselines. We believe these changes address your initial concerns.
>
> We welcome any further questions or the opportunity to discuss any remaining issues. Thank you for your time and consideration.

---

> ### Author Response · Authors · 2023-11-22
>
> Dear Reviewer HpN9,
>
> We submitted our rebuttal eight days ago and sent a reminder 3 days ago, and we would very much appreciate a reply during the reviewer-author discussion, which ends in 14 hours. We would gladly reply promptly if you still have any concerns or questions or replies. We‘re looking forward to your reply. Thank you for your service to ICLR!

---

### Official Review · Reviewer_VW8f · 2023-11-01

**Soundness:** 4 excellent
**Presentation:** 4 excellent
**Contribution:** 4 excellent
**Rating:** 8
**Confidence:** 3

**Summary:**

The paper proposed constrained parameter regularization as an alternative to L2 / weight decay regularization. The paper provided a computationally efficient method, explored different constraints initialization, and delivered experiments on various tasks.

**Strengths:**

1) The methods allow different constraints for different parameter groups which is not possible to achieve with a traditional approach.
2) No computational overhead and the code is open-source.
3) The approach seems to be effective on different empirical tasks e.g. grokking, object detection, and medical image tasks.

**Weaknesses:**

1) It’s difficult to find a good value of the upper bound for each group of the parameters.
2) The accuracy gain for the language model is quite small, so it’s not clear to say that AdaCPR is better than AdamW on this task.
3) See questions

**Questions:**

1) “We can also close the gap between training and validation performance by increasing the weight decay regularization but this comes at the price of unstable training” . Shouldn’t we compare the AdamAdaCPR with AdamW with a bigger regularization parameter? Since our concern here is the grokking effect.

2) There are large performance drops in Table 3 when the number of warm-start steps increases from 3k to 4k. This increment is not large compared to the total number of training steps is 25k. Does this imply that CPR with Kappa-I_s is highly sensitive to the warm-start parameter?

---

> ### Author Response · Authors · 2023-11-14
> **Response to Reviewer VW8f**
>
> We thank the reviewer for the high score and warm words. We address the concerns and questions as follows:
>
> **W1:** *It’s difficult to find a good value of the upper bound for each group of the parameters.*
>
> Indeed, there is still no hyperparameter-free regularization technique. Similar to the weight decay value, one needs to choose e.g. the warm start steps. These could be more  interpretable or transferable than the weight decay value, but we agree that so far this is a weakness and should be addressed in future works.
>
> **W2:** *The accuracy gain for the language model is quite small, so it’s not clear to say that AdaCPR is better than AdamW on this task.*
>
> We now performed three random seeds on the GPT2 experiments and added the standard deviation in the plot and in Table G.6 and G.7 in Appendix G. We also performed a Welch’s t-test on the GPT2s experiments with the best AdamW and best AdamCPR configuration, showing a significant difference with a p-value of 0.018.
>
> **Q1:** *“We can also close the gap between training and validation performance by increasing the weight decay regularization but this comes at the price of unstable training” . Shouldn’t we compare the AdamAdaCPR with AdamW with a bigger regularization parameter? Since our concern here is the grokking effect.*
>
> We agree, we can also compare it to an AdamW training with a higher weight decay, therefore we added a new plot (E.7) in Appendix E. We see the ability of weight decay to close the gap as well. We also see the instability in comparison to a more stable behavior with AdamCPR and AdamAdaCPR.
>
> **Q2:** *There are large performance drops in Table 3 when the number of warm-start steps increases from 3k to 4k. This increment is not large compared to the total number of training steps is 25k. Does this imply that CPR with Kappa-I_s is highly sensitive to the warm-start parameter?*
>
> Thank you for mentioning this issue. The problem arises because the datasets are very small. Consequently, unregulated optimization tends to overfit quickly which leads to NaNs. When CPR regularization starts too late, as was the case after 4000 steps in this instance, the model is already overfitted, resulting in some broken folds. These are counted as zero in the mean, which explains the low number in the table.
>
>  We would be very happy to answer any follow-up questions and concerns.

---

### Author Response · Authors · 2023-11-14
**General response**

We thank the reviewers for their time and valuable feedback. We are glad that the reviewers appreciated our novel approach (R3), flexibility in constraints for different parameter groups (R1), effectiveness across various empirical tasks (R1), provision of open-source code (R1, R2), the tailored approach to regularization, where individual penalties are applied to parameter groups (R2) and robust performance (R3).

We are thankful for the critiques and addressed them by adding new sections and plots in the appendix, performing additional experiments, and adjusting the abstract and manuscript. We are now able to add standard deviations to the image classification and language modeling tasks, allowing us to assess the significance of the results. We added one more related work (AdaDecay, Nakamura & Hong 2019), also as a baseline, along with more baselines on weight decay scheduling. To show the small sensitivity of the update rate, we added a section in the appendix with experiments on different values of $\mu$.

In the direct responses, we will address all questions and weak points individually. We hope we can address your concern and look forward to an interesting discussion.

---

### Meta-Review · Area_Chair_bTDT · 2023-12-11

**Metareview:**

(a) The paper proposes a new parameter regularization method that can vary the weight decay term per each parameters, which is a novel idea. The proposed method does not require additional significant computational budget compared to previous state-of-the-art, AdamW. The effectiveness of the proposed CPR is shown through three different experiments.
(b) Maintaining adaptive decay weight for each parameter is achieved by employing the augmented Lagrangian method effectively.
(c) While the idea seems somewhat novel, the additional gain CPR achieves is quite marginal. Some gains are achieved on smaller datasets (which may benefit from additional regularization), but the experiment on GPT2 shows little difference between AdamW and CPR.  Also, the reviewers were concerned about the stability w.r.t. the warm-start parameter etc.

**Justification For Why Not Higher Score:**

The numerical gain in the experiments compared to baseline is not very significant.

**Justification For Why Not Lower Score:**

N/A

---

### Decision · Program_Chairs · 2024-01-16

Reject